# Transfer is All You Need: Revisiting the Stability–Plasticity Dilemma through Backward and Forward Transfer in PLMs

## Abstract

Incremental Learning (IL) has long been an important research area in neural networks. Since IL requires retaining prior knowledge while learning tasks sequentially, many studies have primarily focused on *'Memory Stability'* to address catastrophic forgetting, while paying less attention to *'Learning Plasticity'*. However, this perspective has recently been challenged. Recent studies have demonstrated that the backbone exhibits sufficiently strong anti-forgetting capabilities, while the classifier (LM Head) is the primary source of forgetting. Moreover, as research on Learning Plasticity has gradually expanded, conflicting findings have emerged regarding the relationship between forgetting and forward transfer. For this issue, we propose a method to evaluate the forgetting and forwarding ability of the backbone itself and compare it with the evaluation in the classifier. To this end, we re-establish the famous metrics BWT (Backward Transfer) and FWT (Forward Transfer) and analyze the correlation between the two. As a result, we find that BWT and FWT are measured completely differently in Classifier, Probing Classifier, and Backbone, and this is the cause of the conflict in previous studies. More specifically, we observed that the considerable capability of the backbone is not effectively transferred to the classifier (LM Head). To address this, we propose *'Just LM-Head Tuning (JLT)'*, a simple yet highly effective approach that leverages the backbone trained through the IL process to transfer the LM Head. JLT is compatible with all existing IL methods and achieves state-of-the-art (SOTA) performance while allowing the backbone to remain unfrozen and continue acquiring knowledge. This effectiveness has been demonstrated not only on older discriminative backbones such as BERT, but also on very recent generative backbones such as LLaMA3.2 and Qwen3 across eight representative benchmarks.

## 1 Introduction

Advances in Artificial Intelligence have come from efforts to mimic the structure of the human brain and way of thinking, both in the structure of the models and in how they learn (Simon, 1981; Dreyfus & Dreyfus, 1991). Humans acquire knowledge incrementally over time, preserve their memories, and cultivate intellect. However, Pretrained Language Models (PLMs), which has been pre-trained with huge parameters and data, has difficulty maintaining performance even in simple sequential fine-tuning. This phenomenon is defined as catastrophic forgetting (French, 1999; Li et al., 2019; Hu et al., 2019; Kaushik et al., 2021; van de Ven et al., 2024), and various studies have been conducted to overcome it (Kirkpatrick et al., 2017; Wang et al., 2023; Yang et al., 2024b).

Incremental Learning (IL) (Polikar et al., 2001; Kemker & Kanan, 2017; Parisi et al., 2019) is a research area that has developed in the direction of preventing previous knowledge from being forgotten while learning new tasks (Parisi et al., 2019). In this field, it has been considered difficult to learn a new task while not forgetting previous memory, resulting in a trade-off relationship, known as the *Stability-Plasticity Dilemma* (Abraham & Robins, 2005; Mermillod et al., 2013; Wu et al., 2021; Araujo et al., 2022). However, existing IL methods have focused somewhat on overcoming catastrophic forgetting rather than finding the optimal point in this dilemma. This is because standard evaluation metrics mainly evaluate the average accuracy for each task, so maintaining previously learned tasks can show better results even if the accuracy of the currently learned task decreases.

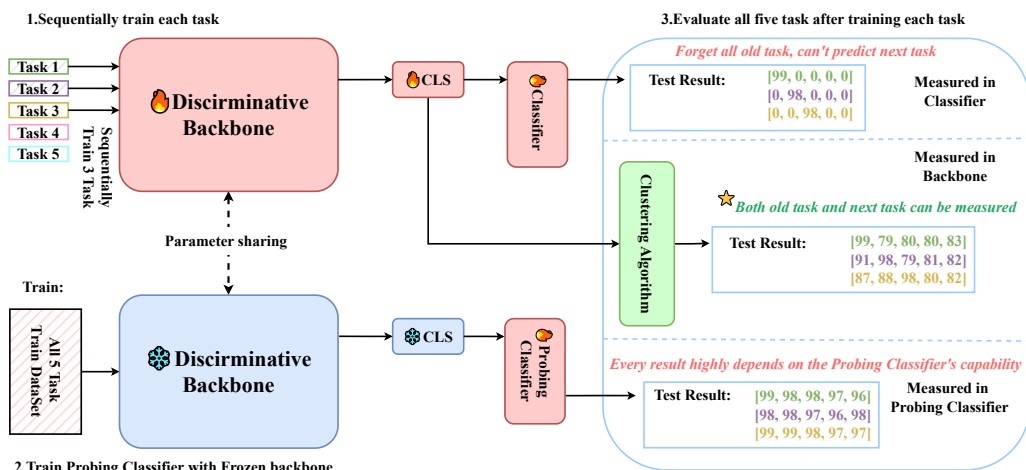

Figure 1: Overview of evaluating BWT and FWT in discriminative backbone. Describes the process of IL and evaluating up to the third task out of five tasks in the CIL scenario. While conducting IL, BWT and FWT are measured by three ways: Classifier, Probing Classifier, and Backbone. In the Generative backbone, Label token are used instead of [CLS] token, and LM Heads take the role of Classifiers. Details are in the Appendix A.

Recent studies have shown that the anti-forgetting ability of the backbone is underestimated, and catastrophic forgetting occurs in the classifier, not the backbone (Davari et al., 2022; Zheng et al., 2024; 2025a). According to the study, the backbone still maintained its performance when evaluated as a separate classifier, and the cause of catastrophic forgetting was that the center of a class already existing in the classifier lost its optimal position during the IL. However, another study introduced a new method for measuring Forward Transfer (FWT) and suggested that less forgetting provides a good inductive bias for FWT (Chen et al., 2023; Zheng et al., 2025b). Then, "Is it possible for the language model to acquire new knowledge while retaining previously learned knowledge?"

We posit that the disruption of existing discourse and the apparent conflicts among prior studies primarily arise from two factors. First, the established evaluation metrics, Backward Transfer (BWT) and FWT, have not functioned as originally intended. To address this limitation, we introduce refined formulations of BWT and FWT that enable more accurate evaluation. Second, the performance outcomes derived from the backbone and from the classifier (LM Head) exhibited substantial heterogeneity. This led to different studies arriving at divergent conclusions depending on the evaluation point. To mitigate this issue, we propose a method for evaluating performance at the backbone and conduct in-depth analysis of BWT and FWT across the classifier, probing classifier, and backbone.

Through the above in-depth and multifaceted analysis, we identified that all existing IL methods fail to sufficiently transfer the strong representational capacity of the backbone to the classifier or LM Head. To address this, we propose a simple yet effective method, Just LM-Head Tuning (JLT), which re-trains only the classifier (LM Head), which is responsible for the model's outputs, on top of the already fully trained backbone. As a result, by applying our method to all existing types of IL approaches (base, replay, knowledge distillation, variational autoencoder), we successfully achieved the upper bound performance of joint fine-tuning. Remarkably, our approach even surpasses recently proposed methods that freeze the backbone—aiming to avoid catastrophic forgetting but, as a consequence, failing to acquire knowledge—while leaving the backbone unfrozen, thereby allowing it to continue acquiring knowledge.

In this study, we validate our proposed method across eight datasets, employing four IL methods with four Transformer encoder backbones and eight IL methods with six Transformer decoder backbones, under both Class-Incremental Learning (CIL) and Task-Incremental Learning (TIL) scenarios. Based on these extensive experiments, we present the contributions of our research as follows.

- Proposing precise definitions of BWT and FWT consistent with their original intent

- Introducing a method to evaluate performance directly at the backbone, independent of the classifier (LM Head)

- Presenting a simple yet effective re-training method for the classifier (LM Head) that can be applied to all IL methods

- Achieving state-of-the-art (SOTA) performance across all benchmarks by combining JLT with existing basic IL methods.

## 2 PROBLEM SETUP AND METRICS

### 2.1 PROBLEM SETUP

IL is defined as follows: To learn a model $f_0 : x \rightarrow y \in Y$ from tasks $D = \{D_1, D_2, \cdots, D_T\}$ and task $D_t = \{(x_t^i, y_t^i)\}_{i=1}$ contains samples $x_t^i \in X_t$ and $y_t^i \in Y_t$. The most commonly studied scenarios in IL are CIL and TIL. In CIL, Classes of different tasks do not overlap: $Y_1 \cap Y_2 \cdots \cap Y_T = \emptyset$. On the other hand, TIL can overlap: $Y_1 \cap Y_2 \cdots \cap Y_T \neq \emptyset$ and you can know which task the class belongs to through task_id. In other words, TIL needs to predict the classes belonging to each task, and CIL needs to predict the classes belonging to all tasks. The CIL scenario, where we evaluate performance on the all task while training each task, is much more challenging than the TIL scenario (Tao et al., 2023), where we only need to maintain performance within each task. then, We discuss the CIL scenario in the main paper, and the TIL scenario in Appendix D.

### 2.2 EVALUATION METRICS FOR IL

**BWT** (Lopez-Paz & Ranzato, 2017; Ebrahimi et al., 2018; 2020) is one of the representative evaluation metrics of IL from the perspective of *'Memory Stability'*, which measures how well the model remembers the tasks it has learned. As shown in Equation 1, it represents the difference between the accuracy immediately after learning the task and the accuracy of the task after learning all tasks (from 1 to the last task $T$).

$$BWT = \frac{1}{T-1} \sum_{i=1}^{T-1} (a_{T,i} - a_{i,i})$$  (1)

where $T$ is the lask task, $a_{T,i}$ is the test accuracy of the $i$-th task of the model trained up to the $T$-th task, and $a_{i,i}$ is the test accuracy of the $i$-th task immediately after training the $i$-th task.

**FWT** is a metric that measures performance from the perspective of *'Learning Plasticity'*, but it has not been used as much as BWT. The biggest reason is that in order to measure how well a new task is learned, '$a_{i-1,i}$' and '$a_{i,i}$' must be compared. But, before learning the $n$-th task, the classifier cannot predict the $n$-th task at all. Therefore, existing studies assume this as random accuracy (Ke et al., 2020; Wołczyk et al., 2021; Ke & Liu, 2022; Wang et al., 2024) or adopt a method of using a separate classifier to evaluate the performance on the $n$-th task in advance Chen et al. (2023). However, these methods depend on the capability of the separate classifier and does not directly participate in IL, which is the main motivation for our research. In conclusion, we measure FWT via Equation 2 which is most consistent with its original intention.

$$FWT = \frac{1}{T-1} \sum_{i=2}^{T} (a_{i,i} - a_{i-1,i})$$  (2)

where $T$ is the lask task, $a_{i,i}$ is the test accuracy of the $i$-th task immediately after learning the $i$-th task, and $a_{i-1,i}$ is the test accuracy of the $i$-th task of the model that learned up to the $i - 1$-th task.

## 3 EXPERIMENTAL SETUP

### 3.1 TASK & BASELINE

We perform five widely used sentence-level tasks for NLP IL on seven datasets, as shown in Table 3. For text classification, we use three datasets, AGNews, DBPedia, and YaHoo, as Topic3Datasets (Zhang et al., 2015). For intent classification, we use CLINC150 (Larson et al.,

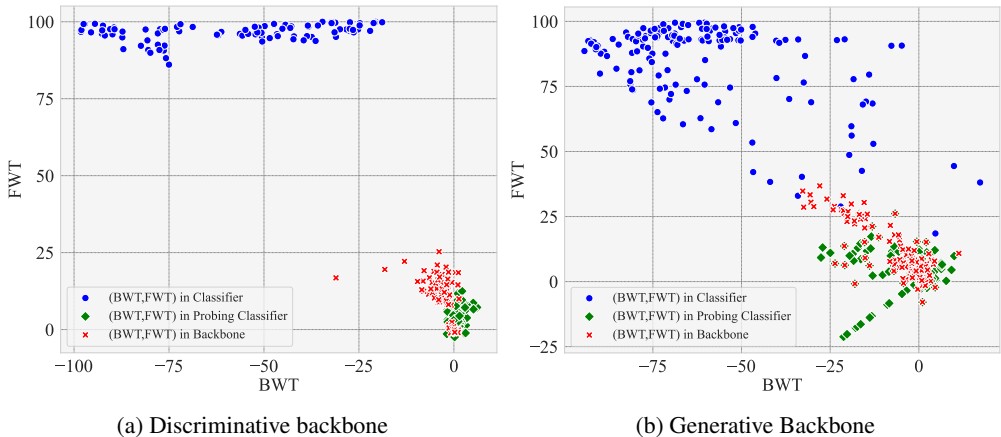

(a) Discriminative backbone          (b) Generative Backbone

Figure 2: BWT and FWT evaluated on four discriminative backbones (five tasks, four IL methods) and six generative backbones (five tasks, eight IL methods).

2019) and Banking77 (Casanueva et al., 2020). Finally, for relation extraction, we use FewRel (Han et al., 2018) and TACRED (Zhang et al., 2017).

For discriminative backbone, we adopt four representative IL methods: Base, ER (Chaudhry et al., 2019), DER++ (Buzzega et al., 2020), and CLSER (Arani et al., 2022). For generative backbone, we compare a total of eight baselines: L2KD Chuang et al. (2020), LAMOL_g, LAMOL_t (Sun et al., 2020), LAMOL_KD (Zheng et al., 2024), and PCLL (Zhao et al., 2022), which were attempted in the generative backbone and Base, DERpp, CLSER. We are aware of recent SOTA(State-Of-The-Art) methods, such as SEQ* (Zheng et al., 2024) and KLDA (Momeni et al., 2025), that freeze their backbones and do not update them. However, since these methods do not update their backbones, their BWT and FWT values are both 0, and thus they are not included in the backbone evaluation. For detailed descriptions of the baselines, please refer to the Appendix C.

## 3.2 BACKBONE

We adopt both discriminative backbones (Encoder architecture backbones) and generative Backbones (Decoder architecture backbones). We adopt the following model to evaluate all baselines. For discriminative backbones, we use BERT-base, BERT-large (Devlin et al., 2019), RoBERTa-base, and RoBERTa-large (Liu et al., 2019b), and for generative backbones, we use Pythia (Biderman et al., 2023) models based on GPT-NeoX (Black et al., 2022) in different sizes (70m, 160m, 410m) and Qwen2-0.5B (Yang et al., 2024a), Qwen2.5-0.5B (Qwen et al., 2025), Qwen3-0.6B (Yang et al., 2025). Due to resource limitations, we cannot experiment with all baselines, but the following models were tested using only Base method to evaluate them according to model type and size. Pythia-1.4B, 2.8B, 6.9B with GPT as the base model. Llama3.2-1B, 3B, Llama3.1-8B, which use Llama as the base model (Dubey et al., 2024); and Qwen3-0.6B, 1.7B, 4B, 8B, which use Qwen as the base model. More details on training and evaluation in Backbone are in the Appendix A.1.

## 4 OBSERVATION: EVALUATION BY THREE DIFFERENT METHOD

### 4.1 BWT & FWT IN CLASSIFER

The most basic evaluation method is to evaluate the output of a classifier that is learned sequentially along with the model. According to previous research, the classifier does not remember previous tasks in sequential learning and predicts all inputs only as recently learned tasks (Hou et al., 2019; Wu et al., 2019; 2022; Zheng et al., 2024). Therefore, the catastrophic forgetting occurs, where the accuracy of the previous task measured by the classifier in the Base method (just sequentially fine-tuning) all becomes 0. In Figure 2a, BWT of the Blue Point shows a large difference by each IL method. It seems that the IL research succeeded in remembering the previous task, but it is unclear whether this prevents forgetting of the backbone or the classifier. In evaluations at the classifier,

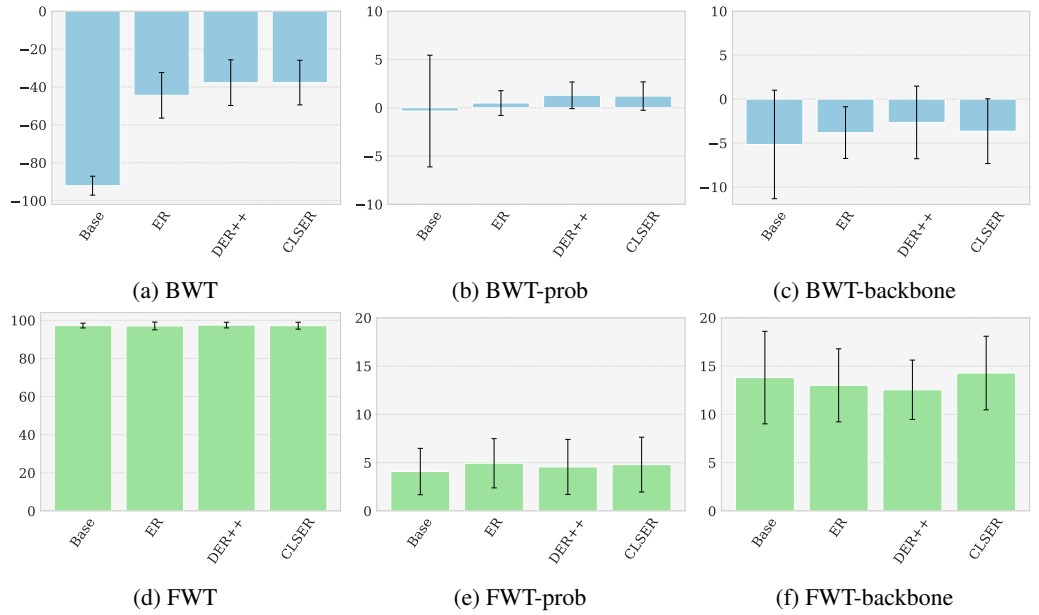

Figure 3: Average and standard deviation of BWT, FWT, BWT-prob, FWT-prob, BWT-backbone, FWT-backbone for each IL method in four discriminative backbones and five tasks. Across both models and tasks, a strong negative correlation between BWT and FWT was consistently observed for each IL method.

FWT presents a more difficult problem. Since the classifier cannot predict unseen tasks, FWT cannot be meaningfully measured there. As shown in Figure 2a, the Blue Point exhibits uniformly high FWT values, exceeding 80 across all models and IL methods.

## 4.2 BWT & FWT in Probing Classifer

In previous research, as a way to prove that the anti-forgetting ability of the backbone is underestimated, a separate classifier that does not participate in IL is used, as shown in Figure 1. This measurement method demonstrates the robustness of the backbone's anti-forgetting ability by training a new classifier at every single task. This approach avoids the bias and forgetting of the existing classifier (Hou et al., 2019; Zhou et al., 2022; Zheng et al., 2024). However, ironically, this relies heavily on the performance of the new classifier, which does not measure forgetting correctly.

In Figure 2a, the BWT of the Green shows a value close to 0 for all models and IL methods. This is because regardless of how many tasks the backbone learned, it always maintained high performance for all tasks due to the outstanding ability of the Probing Classifier. However, according to a previous study (Zhou & Srikumar, 2021b) that analyzed the backbone representation and the classifier separately, the classifier can achieve excellent performance even if the backbone representation is somewhat insufficient, which implies that there is a limit to evaluating the backbone itself.

As presented in Figure 1, FWT through a probing classifier freezes the backbone after each task is learned and then trains the probing classifier across all tasks. Therefore, it becomes possible to predict the entire class with the features of the backbone that learned each task, and it can be evaluated according to the definition of FWT in Section 2.2. However, in this case, it shows high performance without discrimination for all tasks by simply learning about one task. In the probing classifier, BWT and FWT seem to have no backward and forward transfer during the IL process.

## 4.3 BWT & FWT in Backbone

To overcome the limitations of existing evaluation methods, we newly evaluate the BWT and FWT of the backbone using a very traditional clustering algorithm. Simply, as presented in Figure 1, we measure BWT-backbone and FWT-backbone using the clustering algorithm for the representations

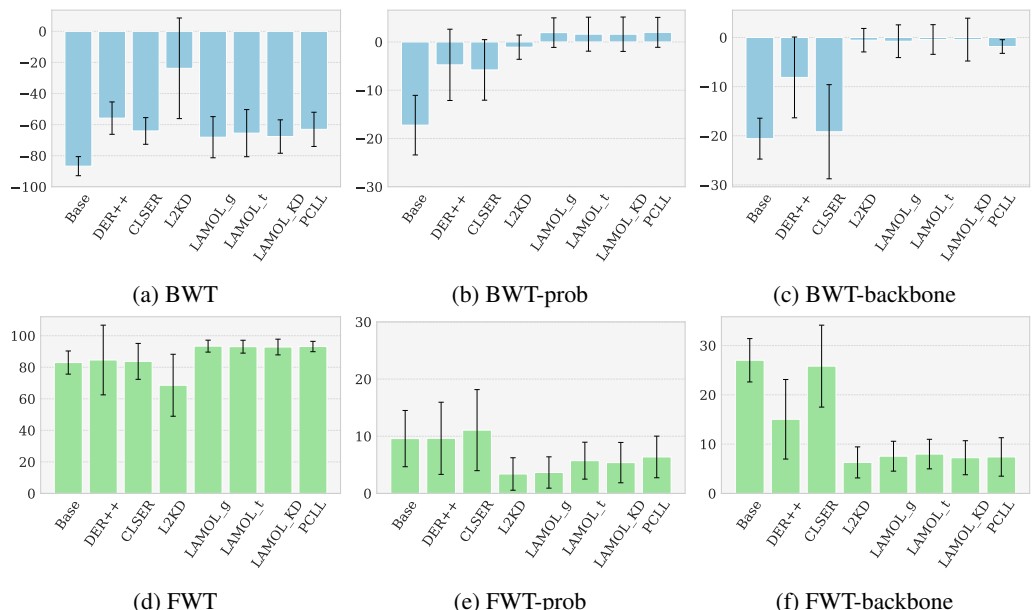

Figure 4: Average and standard deviation of BWT, FWT, BWT-prob, FWT-prob, BWT-backbone, FWT-backbone for each IL method in six generative backbones and five tasks. Across both models and tasks, a strong negative correlation between BWT and FWT was consistently observed for each IL method.

of the backbone. We propose this as an 'Auxiliary Evaluation' to directly measure the BWT and FWT of the backbone, which are otherwise difficult to assess due to catastrophic forgetting and strong bias in the classifier. There has been a recent attempt to use the diversity of backbone features as a means of 'Auxiliary Evaluation' of IL (Chen et al., 2023). Many studies have already attempted to analyze using backbone's representations to avoid dependence on or bias in classifiers, and have shown performance that is not significantly inferior to that of using a classifier (Liu et al., 2019a; Zhou & Srikumar, 2021a;b).

### 4.3.1 CLUSTERING ALGORITHM & EVALUATION METRICS

To evaluate the features extracted from the backbone, we employed five representative clustering algorithms: K-means (MacQueen, 1967), Gaussian Mixture Model (Dempster et al., 1977), Spectral Clustering (Ng et al., 2001), Agglomerative Clustering (Johnson, 1967), and Deep Clustering (Xie et al., 2016). While all algorithms were evaluated, our main analysis and results are based on Spectral Clustering. Further details are provided in Appendix B.

For the evaluation, we cluster the backbone features by 'number of classes', map the results to actual labels, and evaluate them with three metrics: ACC (Accuracy), ARI (Adjusted Rand Index), and NMI (Normalized Mutual Information) (Hubert & Arabie, 1985; Strehl & Ghosh, 2002; Vinh et al., 2009). In TIL scenario, where only a small number of classes in the task need to be evaluated, Acc, which simply maps the clustering results to the actual labels, is sufficient for evaluation. However, in CIL scenario where many classes must be classified, the reliability of Acc is lowered, and it is evaluated through ARI and NMI, which are metrics to complement this (Zhang et al., 2019). Details on the evaluation metrics are in the Appendix B.1.

## 5 OBSERVATION RESULTS

### 5.1 DISCRIMINATIVE BACKBONE

In Figure 3, BWT-backbone shows a little bit of forgetting, which is consistent with previous studies that the anti-forgetting ability of backbone is robust (Zheng et al., 2024). BWT-backbone is clearly

| | Scenario:CIL | Discriminative Backbone | | | | Generative Backbone | | | | | |
|---|---|---|---|---|---|---|---|---|---|---|---|
| | Metric | BERT-b | BERT-l | RoBERTa-b | RoBERTa-l | Pythia-70m | Pythia-160m | Pythia-410m | Qwen2-0.5B | Qwen2.5-0.5B | Qwen3-0.6B |
| Classifier | Acc | 0.4272 | 0.1398 | 0.3807 | 0.2429 | -0.5039 | -0.3631 | -0.3738 | -0.5201 | -0.3624 | -0.3790 |
| Prob Classifier | Acc | 0.6478 | 0.1130 | 0.2468 | 0.1079 | -0.4180 | -0.4845 | -0.6243 | -0.4312 | -0.4823 | -0.6187 |
| K-means | Acc | -0.6281 | -0.5621 | -0.4811 | -0.5233 | -0.8463 | -0.8721 | -0.8513 | -0.8520 | -0.8698 | -0.8576 |
| | ARI | -0.6638 | -0.5132 | -0.5314 | -0.5419 | -0.7992 | -0.8103 | -0.8192 | -0.8021 | -0.8135 | -0.8250 |
| | NMI | -0.6798 | -0.6120 | -0.5822 | -0.6788 | -0.8193 | -0.9147 | -0.8921 | -0.8222 | -0.9080 | -0.8953 |
| GMM | Acc | -0.5183 | -0.5712 | -0.4956 | -0.4278 | -0.7689 | -0.7493 | -0.7742 | -0.7705 | -0.7512 | -0.7790 |
| | ARI | -0.6762 | -0.6888 | -0.6232 | -0.5145 | -0.8101 | -0.8018 | -0.8439 | -0.8124 | -0.8040 | -0.8471 |
| | NMI | -0.6886 | -0.6982 | -0.6123 | -0.6121 | -0.7914 | -0.8327 | -0.8431 | -0.7933 | -0.8354 | -0.8460 |
| Spectral | Acc | -0.6918 | -0.6128 | -0.6157 | -0.5987 | -0.8102 | -0.8333 | -0.8688 | -0.8118 | -0.8305 | -0.8650 |
| | ARI | -0.6841 | -0.6233 | -0.6822 | -0.6522 | -0.8239 | -0.8484 | -0.8129 | -0.8260 | -0.8452 | -0.8154 |
| | NMI | -0.7213 | -0.6434 | -0.6557 | -0.6857 | -0.8231 | -0.8923 | -0.8725 | -0.8257 | -0.8894 | -0.8699 |
| Agglomerative | Acc | -0.4744 | -0.5114 | -0.6144 | -0.4566 | -0.7718 | -0.7466 | -0.6999 | -0.7740 | -0.7481 | -0.7020 |
| | ARI | -0.7413 | -0.6912 | -0.5989 | -0.7362 | -0.8654 | -0.8132 | -0.8948 | -0.8623 | -0.8180 | -0.8905 |
| | NMI | -0.9091 | -0.7122 | -0.7321 | -0.7487 | -0.8835 | -0.9263 | -0.8726 | -0.8810 | -0.9230 | -0.8752 |
| Deep Clustering | Acc | -0.5002 | -0.4872 | -0.6166 | -0.5871 | -0.7943 | -0.8221 | -0.7849 | -0.7971 | -0.8195 | -0.8802 |
| | ARI | -0.4748 | -0.5237 | -0.5891 | -0.4824 | -0.8412 | -0.8109 | -0.8019 | -0.8390 | -0.8137 | -0.8065 |
| | NMI | -0.6258 | -0.6413 | -0.6709 | -0.6235 | -0.8741 | -0.8323 | -0.8824 | -0.8720 | -0.8350 | -0.8922 |

Table 1: Pearson correlation coefficient between BWT and FWT for each metric in CIL scenario.

different from BWT, which was considered to forget all previous tasks due to the classifier, and also different from BWT-prob, which measured that forgetting of the backbone did not occur at all, even at the base method. Moreover, FWT evaluation shows a more clear difference from the two evaluation methods. While all FWT values exceed 80 and FWT-prob remains close to 0, FWT-backbone ranges between 0 and 25, indicating that the backbone retains a certain level of performance for the next task, which further improves after training. In Table 1, BWT-backbone and FWT-backbone show a strong negative pearson correlation, indicating a trade-off relationship of forwarding as much as forgetting, which is in line with the Stability-Plasticity Dilemma. On the other hand, BWT and FWT, and BWT-prob and FWT-prob even show positive correlation rather than negative.

## 5.2 GENERATIVE BACKBONE

In Figure 2b, 4, BWT, BWT-prob, and BWT-backbone all show relatively wider ranges than the discriminative backbone. Even in this case, BWT showed results in which forgetting occurred significantly, with many experiments scoring below -80. On the other hand, BWT-backbone shows that the anti-forgetting ability of the backbone is much better than that of LM head, just like in the discriminative backbone. However, in Table 1, all three evaluation methods in the generative backbone have stronger negative correlations than in the discriminative backbone. Among them, BWT-backbone and FWT-backbone exhibit a strong negative correlation, with coefficients even below -0.9. The results of the Spearman correlation are in Table 7. From experimental results, we draw the following conclusions:

- Unlike the classifier, which suffers from severe catastrophic forgetting, the backbone exhibits strong anti-forgetting capability even under the base method.
- The Stability–Plasticity Dilemma remains strongly valid at the backbone, but not at the classifier (LM head).
- The capability of the backbone is not sufficiently transferred to the classifier (LM Head).

## 6 JUST LM-HEAD TUNING

We propose a simple yet intuitive approach to ensure that the sufficient capability of the backbone can be effectively reflected in the final output through the LM head. Specifically, we introduce Just LM-Head Tuning (JLT), in which the LM head is lightly trained on the backbone's representations using just the same training data after any IL method has been applied. This method does not require additional architectural components such as adapters, encoders, or classifiers, nor does it enforce parameter freezing of the backbone to artificially restrict updates. After training the model with each IL method, we fine-tune only the LM Head parameters $W \in \mathbb{R}^{V \times d}$ based on the backbone outputs, where $V$ is the vocabulary size and $d$ is the hidden dimension. At the $n$-th task, the dataset is defined as (as defined in Section 2.1)

$$D_n = \{(x_n^i, y_n^i)\}_{i=1}^{N_n}, \quad x_n^i \in X_n, \ y_n^i \in Y_n,$$

Figure 5: Overview of JLT. The IL process follows the basic procedure of each IL method, and after all IL training is completed, the LM Head is trained using the representation of the backbone. Only the transfer of the backbone's capability to the LM Head is performed.

| Model | IL Method | Tacred $\mathcal{A}_t$ | BWT | FWT | Banking77 $\mathcal{A}_t$ | BWT | FWT | Clinc150 $\mathcal{A}_t$ | BWT | FWT | Fewrel $\mathcal{A}_t$ | BWT | FWT | Topic3 $\mathcal{A}_t$ | BWT | FWT |
|---|---|---|---|---|---|---|---|---|---|---|---|---|---|---|---|---|
| GPT2-NEOX (Pythia) 410M | Joint Fine-tuning | 98.99 | - | - | 95.66 | - | - | 96.33 | - | - | 95.35 | - | - | 95.50 | - | - |
| | SEQ* | 44.34 | 0.00 | 0.00 | 67.12 | 0.00 | 0.00 | 84.51 | 0.00 | 0.00 | 61.99 | 0.00 | 0.00 | 70.56 | 0.00 | 0.00 |
| | KLDA-E | 97.20 | 0.00 | 0.00 | 93.03 | 0.00 | 0.00 | 96.62 | 0.00 | 0.00 | 94.55 | 0.00 | 0.00 | 94.53 | 0.00 | 0.00 |
| | Base | 11.97 | -91.26 | 91.28 | 7.56 | -75.42 | 68.86 | 4.67 | -89.69 | 87.62 | 6.17 | -80.57 | 73.59 | 20.53 | -91.02 | 93.08 |
| | L2KD | 24.68 | -79.27 | 94.39 | 52.32 | -35.08 | 85.72 | 26.67 | -73.17 | 94.74 | 30.18 | -53.23 | 74.57 | 58.17 | 10.27 | 40.25 |
| | LAMOL_KD | 32.81 | -69.55 | 93.68 | 49.48 | -54.92 | 97.08 | 42.09 | -60.48 | 97.50 | 27.70 | -75.60 | 95.43 | 50.08 | -54.98 | 92.97 |
| | PCLL | 24.30 | -65.29 | 74.37 | 45.91 | -56.97 | 84.41 | 44.16 | -57.02 | 91.81 | 29.79 | -68.41 | 82.84 | 55.80 | -42.23 | 92.66 |
| | Base + JLT | 98.93 | -0.55 | 11.38 | 75.07 | -19.92 | 84.58 | 72.47 | -25.50 | 92.29 | 51.31 | -36.10 | 77.50 | 72.21 | -14.80 | 79.92 |
| | L2KD + JLT | 98.04 | -0.25 | 4.92 | 95.23 | -2.99 | 88.56 | 96.20 | -3.38 | 94.52 | 95.18 | -1.04 | 93.52 | 93.91 | -0.16 | 92.95 |
| | LAMOL_KD + JLT | 98.24 | 0.25 | 9.59 | 95.03 | -3.33 | 88.98 | 96.38 | -2.95 | 93.40 | 95.73 | -0.54 | 92.82 | 94.00 | -0.08 | 92.99 |
| | PCLL + JLT | 96.70 | -0.58 | 10.29 | 93.31 | -3.60 | 87.48 | 96.27 | -3.29 | 91.36 | 94.70 | -1.06 | 92.15 | 93.46 | -0.26 | 93.51 |
| LLaMA3.2 1B | Joint Fine-tuning | 98.33 | - | - | 96.26 | - | - | 97.33 | - | - | 95.65 | - | - | 95.50 | - | - |
| | SEQ* | 45.82 | 0.00 | 0.00 | 67.48 | 0.00 | 0.00 | 83.72 | 0.00 | 0.00 | 62.13 | 0.00 | 0.00 | 71.64 | 0.00 | 0.00 |
| | KLDA-E | 97.36 | 0.00 | 0.00 | 93.27 | 0.00 | 0.00 | 96.71 | 0.00 | 0.00 | 94.68 | 0.00 | 0.00 | 94.59 | 0.00 | 0.00 |
| | Base | 12.23 | -91.60 | 92.01 | 11.56 | -91.10 | 89.28 | 5.89 | -93.21 | 92.45 | 8.42 | -81.39 | 77.03 | 19.50 | -88.74 | 97.84 |
| | L2KD | 31.83 | -68.77 | 81.20 | 51.56 | -32.46 | 76.52 | 26.44 | -66.62 | 87.88 | 32.75 | -54.67 | 78.49 | 63.17 | 17.27 | 38.09 |
| | LAMOL_KD | 32.75 | -62.04 | 86.04 | 45.85 | -64.91 | 87.01 | 44.80 | -54.05 | 92.93 | 29.96 | -71.79 | 89.10 | 48.10 | -56.19 | 92.72 |
| | PCLL | 31.18 | -77.22 | 93.48 | 49.03 | -67.69 | 87.40 | 40.69 | -61.93 | 93.45 | 28.46 | -74.38 | 93.88 | 48.84 | -55.42 | 92.89 |
| | Base + JLT | 98.63 | -0.74 | 21.56 | 79.47 | -17.08 | 87.05 | 74.91 | -23.33 | 93.45 | 51.29 | -36.30 | 79.57 | 73.06 | -13.23 | 79.85 |
| | L2KD + JLT | 97.06 | -1.76 | 17.26 | 96.10 | -2.20 | 87.46 | 96.36 | -2.95 | 95.02 | 95.35 | -1.00 | 94.38 | 93.82 | -0.16 | 92.79 |
| | LAMOL_KD + JLT | 97.15 | -0.83 | 8.44 | 95.36 | -2.42 | 84.13 | 87.84 | -8.10 | 89.05 | 94.72 | -0.99 | 92.16 | 90.68 | -1.03 | 89.74 |
| | PCLL + JLT | 96.30 | -0.32 | 18.01 | 94.32 | -2.54 | 87.33 | 96.29 | -3.29 | 91.43 | 95.42 | -0.97 | 94.94 | 92.24 | -0.10 | 91.98 |
| Qwen3 0.6B | Joint Fine-tuning | 98.66 | - | - | 93.36 | - | - | 96.66 | - | - | 95.65 | - | - | 96.50 | - | - |
| | SEQ* | 44.09 | 0.00 | 0.00 | 66.84 | 0.00 | 0.00 | 82.91 | 0.00 | 0.00 | 60.77 | 0.00 | 0.00 | 70.41 | 0.00 | 0.00 |
| | KLDA-E | 96.94 | 0.00 | 0.00 | 92.81 | 0.00 | 0.00 | 96.25 | 0.00 | 0.00 | 94.33 | 0.00 | 0.00 | 94.21 | 0.00 | 0.00 |
| | Base | 11.30 | -90.94 | 90.14 | 12.73 | -92.39 | 91.44 | 6.60 | -90.57 | 90.64 | 6.98 | -79.11 | 73.01 | 19.99 | -89.91 | 89.94 |
| | L2KD | 34.67 | -67.41 | 90.38 | 55.78 | -25.99 | 75.11 | 27.31 | -63.83 | 86.02 | 32.90 | -58.13 | 73.79 | 62.32 | 9.90 | 44.41 |
| | LAMOL_KD | 42.20 | -59.61 | 94.17 | 52.44 | -52.31 | 97.42 | 42.71 | -59.83 | 93.07 | 26.45 | -79.12 | 95.30 | 42.09 | -60.48 | 93.50 |
| | PCLL | 49.40 | -55.33 | 93.50 | 52.99 | -50.68 | 97.78 | 43.27 | -59.81 | 90.10 | 31.29 | -74.50 | 96.14 | 49.40 | -55.33 | 92.50 |
| | Base + JLT | 97.49 | -1.73 | 15.76 | 75.26 | -22.58 | 86.36 | 77.24 | -21.31 | 93.24 | 52.55 | -35.27 | 78.87 | 71.03 | -18.32 | 82.20 |
| | L2KD + JLT | 98.21 | -0.15 | 10.69 | 94.93 | -3.67 | 86.40 | 95.27 | -4.40 | 93.12 | 95.58 | -1.15 | 94.98 | 92.16 | -1.21 | 91.32 |
| | LAMOL_KD + JLT | 97.88 | -0.53 | 13.95 | 92.08 | -6.82 | 84.89 | 90.47 | -9.50 | 92.07 | 95.01 | -1.87 | 94.67 | 91.47 | -1.53 | 90.21 |
| | PCLL + JLT | 96.23 | -0.07 | 17.69 | 92.66 | -5.00 | 87.93 | 95.22 | -4.36 | 92.36 | 94.96 | -1.13 | 96.55 | 89.56 | -1.75 | 90.11 |

Table 2: Experimental results of combining representative types of IL methods with JLT on three generative models. Joint Fine-tuning denotes fine-tuning on all tasks simultaneously. $\mathcal{A}_t$ is defined according to Equation 3, BWT according to Equation 1, and FWT according to Equation 2.

with disjoint label sets $Y_1, \ldots, Y_T$ in the CIL setting ($Y_1 \cap Y_2 \cap \cdots \cap Y_T = \emptyset$). Let $\mathcal{Y}_{1:n} = Y_1 \cup \cdots \cup Y_n$ denote the union of all classes observed so far.

For each input $x_n^i$, the backbone produces a representation

$$h_n^i = f_\theta(x_n^i) \in \mathbb{R}^d,$$

and the LM Head maps it to vocabulary logits

$$z_n^i = W h_n^i, \qquad p_n^i = \text{softmax}(z_n^i).$$

Each class $y \in \mathcal{Y}_{1:n}$ is associated with a representative token $\tau(y) \in \{1, \ldots, V\}$, and the training objective at task $n$ is

$$\mathcal{L}_n = -\frac{1}{N_n} \sum_{i=1}^{N_n} \log p_n^i [\tau(y_n^i)],$$

which requires discriminating among all classes in $\mathcal{Y}_{1:n}$. All experiments were conducted three times for each IL method, and the average values of all metrics are presented. The implementation details of JLT can be found in the Appendix F. The distinguishing characteristics and advantages of JLT, compared to existing classifier-learning methods or approaches that freeze the backbone, are as follows:

- JLT does not use any independent or additional structures.

- The backbone is not frozen and thus fully undergoes all IL training and acquires knowledge.

- JLT can be combined with any IL method in a plug-and-play manner.

We integrated JLT with representative IL methods, namely Base, L2KD (old sample replay), LAMOL_KD (Knowledge Distillation), and PCLL (variational autoencoder). Since JLT requires tuning only the LM Head on top of the backbone after the entire process, it exhibits broad applicability and scalability across all IL methods. We present in Table 2 the experimental results obtained by combining the four representative IL methods with three widely used open-source LLMs architectures (GPT, LLaMA, and Qwen) across all benchmarks. Remarkably, substantial performance improvements were observed across all benchmarks. In particular, even in the case of the simple Base + JLT, which merely performs sequential fine-tuning, significant performance gains were achieved on all models and benchmarks. This finding is consistent with prior work and the results shown in Figure 4c. The backbone already demonstrated strong anti-forgetting ability, and the phenomenon of catastrophic forgetting, long considered to imply total information loss, was revealed to originate not from the backbone itself but rather from the LM Head during the learning process.

A closer observation reveals that, for the Tacred benchmark, all models already exhibited sufficient performance on tasks that had not been explicitly trained. This is reflected in the FWT values of the methods combined with JLT, which remained relatively small due to the models' already strong performance. In contrast, for the other four benchmarks, large FWT values were observed, indicating that the models initially possessed little to no competence on those tasks. While the conventional FWT metric could not capture such phenomena, our formulation in Equation 2 enabled us to measure and present performance levels before learning the task. These experimental results suggest that LLMs, having been pretrained on massive amounts of data, may already possess substantial capability on certain benchmarks. Notably, on the Tacred benchmark, all methods combined with JLT achieved an $\mathcal{A}_t$ score exceeding 96, with BWT values approaching zero.

When examining the differences across the four methods, only Base + JLT exhibited a certain degree of information loss. This result is consistent with the findings in Figure 4c. On average, the Base method recorded a BWT of around -20, whereas L2KD, LAMOL_KD, and PCLL achieved BWT values close to 0. These results closely align with the outcomes in the Table 2 for the methods with JLT applied. This demonstrates that the approach we proposed in Section 4.3 for evaluating BWT and FWT in the backbone is highly effective. Except for Base + JLT, the methods applying JLT to L2KD, LAMOL_KD, and PCLL all achieved performance very close to Joint Fine-tuning, which is regarded as the upper bound. This confirms the effectiveness of applying replay, knowledge distillation, and autoencoder techniques to IL research. Furthermore, it shows that JLT can exert a strong effect across all types of IL methods.

## 7 CONCLUSION

We started our research motivated by the fact that most studies in IL only evaluate performance based on average accuracy at the classifier. Even existing FWT evaluations did not reflect the intended purpose of FWT, leading to conflicting findings regarding the Stability–Plasticity Dilemma. Through our backbone evaluation method, we were able to measure FWT in accordance with its intended purpose, and also measure BWT while avoiding the catastrophic forgetting effects of the classifier. With this, we resolved the conflicts among prior studies and theories, and further revealed the backbone's inherent anti-forgetting and forward transfer capabilities. Based on these observations, we propose Just LM-Head Tuning (JLT), a method designed to effectively transfer the sufficient capability of the backbone to the LM Head. JLT is compatible with all existing IL methods while achieving state-of-the-art (SOTA) performance. We believe that our findings can be broadly applied to future IL research, from the design of new methods to the evaluation process.

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

## LIMITATIONS

We use Clustering Algorithm to measure forgetting and forwarding in backbone, but we know that it may not be as complete as evaluating through classifier. Our study is not to propose a perfect new evaluation method, but to observe the change in IL process of backbone. Incremental learning tasks that require sequential fine-tuning of models require a lot of resources, which limits experiments with larger generative models. We tried to use Qwen3-32B Yang et al. (2025), Llama3-70B Dubey et al. (2024), etc. as generative models, but there were resource limitations and difficulties in applying each IL method equally. Large models larger than 8B must be learned using PEFT Houlsby et al. (2019); Hu et al. (2021); Han et al. (2024) due to resource limitations, but in this case, there were IL methods (L2KD, LAMOL, LAMOL_KD) that did not work. Our resources were limited to training models less than 10B, and thus it was impossible to measure BWT and FWT according to differences in model sizes. We measured the BWT, FWT of the backbone and experimentally showed that BWT+FWT is almost constant, but we did not propose an IL method to solve this, and proposing a new IL method will be a future study.

## A EXPERIMENTAL DETAILS

### A.1 BACKBONE DETAILS

For the discriminative backbone, learning and evaluation are performed using the [CLS] token features of the last hidden states (Ethayarajh, 2019). At this time, the classifier is a linear layer that uses the output dimension of the backbone as the input dimension and the number of classes for each task as the output dimension. In the TIL task, the task is learned and evaluated from the classifier as is, and in the CIL task, the logits of the classifier for each task are concatenated and used. To use the generative backbone in sentence-level tasks, we use two types of prompts depending on the task type. For text and intent classification, we use the following prompt:

```
"Input sentence: {text}\n The Label: {label}{eos token}"
```

For relation extraction, we use the following prompt:

```
"Input sentence: {text}\n The relationship between {head entity}
and {tail entity} is {label}{eos token}"
```

With the above prompts, we use causal language modeling loss to optimize labeleos token (Zheng et al., 2024). However, depending on the IL method, we additionally use Cross-Entropy loss, KL-Divergence loss, MSE loss, etc.

### A.2 IMPLEMENTATION DETAILS

We use the following settings for the five tasks. For Topic3Datasets, we applied a max len of 256 and 3 epochs for each incremental task, for FewRel and TACRED, we applied a max len of 128 and 5 epochs, and for CLINC150 and Banking77, we applied a max len of 64 and 5 epochs. We used a learning rate of $1 \times 10^{-5}$ for each backbone and $1 \times 10^{-3}$ for the classifier with the AdamW optimizer (Kinga et al., 2015). When fine-tuning a probing classifier with the frozen features of the backbone, we train for 20 epochs, and all classifiers use the logit of the linear layer.

We also used four NVIDIA RTX 3080 (VRAM 24G) and eight NVIDIA A5000 (VRAM 24G) for our experiments. All experiments were performed three times, and the average values were used for visualization. Other than that, we used the basic parameter settings recommended in each study for the incremental learning methods.

### A.3 EVALUATION METRICS

#### A.3.1 AVERAGE ACCURACY

The most basic method to evaluate Incremental Learning is to measure the performance of all tasks after learning the final task. Accordingly, Average Accuracy has been recognized as the best IL method, and has been used as a representative metric in almost all studies. However, when learning

| Task | Dataset | # Classes | # Tasks | # CIL Classes | # TIL Classes | # Training Instances | # Test Instances |
|---|---|---|---|---|---|---|---|
| Text Classification | Topic3Datasets | 25 | 5 | 25 | 5 | 75000 | 46000 |
| Intent Classification | CLINC150 | 150 | 15 | 150 | 10 | 15000 | 4500 |
| | Banking77 | 77 | 7 | 77 | 11 | 7191 | 2800 |
| Relation Extraction | FewRel | 80 | 8 | 80 | 10 | 33600 | 11200 |
| | TACRED | 40 | 8 | 40 | 5 | 5909 | 1259 |

Table 3: The statistics on selected sentence-level datasets for IL. **Tasks** is the number of incremental tasks for each dataset, **CIL Classes** is the number of test set classes in the CIL scenario, and **TIL Classes** is the number of test set classes evaluated for each task in the TIL scenario.

and evaluating a total of $N$ tasks, it is much more advantageous to maintain the performance of $N-1$ previous tasks than the performance of the current 1 task, so there is an aspect that IL methods focus excessively on anti-forgetting. Average Accuracy in task $t$ is defined as follows:

$$\mathcal{A}_t = \frac{1}{t} \sum_{i=1}^{t} a_{t,i} \tag{3}$$

where $a_{t,i}$ represents the accuracy of the model incrementally learned from task 1 to $t$ on the test set of task $i$.

### A.3.2 AVERAGE INCREMENTAL ACCURACY

Average Incremental Accuracy calculates the average of Average Accuracy from 1 to the last $T$ task. This metric can prevent performance from being evaluated only by $A_T$, but overall, it shows a similar trend to Average Accuracy. Since the previous task accuracy still has a high proportion, maintaining the accuracy of the previous task is good for the overall result.

$$\overline{\mathcal{A}} = \frac{1}{T} \sum_{t=1}^{T} A_t \tag{4}$$

## B CLUSTERING ALGORITHM DETAILS

### B.1 CLUSTERING EVALUATION METRICS

**ACC (Accuracy)**

A contingency matrix is created to map the results from each clustering algorithm to the actual labels. The Hungarian algorithm is used on the contingency matrix to find the optimal mapping with the actual labels and evaluate the performance Hubert & Arabie (1985).

**ARI (Adjusted Rand Index)**

ARI measures the agreement between the clustering results and the actual labels, and compensates for the random prediction results. ARI is calculated based on pairs of data points. It compares the correspondence between two clusterings for each pair of data points. It has a scale between -1 and 1, where 1 indicates perfect agreement, 0 indicates agreement at the level of random clustering, and -1 indicates agreement lower than expected (random clustering) (Vinh et al., 2009; Zhang et al., 2019).

$$ARI = \frac{2 \cdot (TP \cdot TN - FN \cdot FP)}{(TP+FN) \cdot (FN+TN) + (TP+FP) \cdot (FP+TN)} \tag{5}$$

**TP(True Positive)**: The number of sample pairs that belong to the same cluster in both clusterings.

**TN(True Negative)**: The number of sample pairs that belong to different clusters in both clusterings.

**FP(False Positive)**: The number of sample pairs that belong to the same cluster in one clustering, but to different clusters in the other clustering.

**FN(False Negative)**: The number of sample pairs that belong to different clusters in one clustering, but to the same cluster in the other clustering.

| Scenario:CIL | | Task | | | | |
|---|---|---|---|---|---|---|
| Clustering Algorithm | Metric | Tacred | Banking77 | Clinc150 | Fewrel | Topic3 |
| K-means | Acc | 52.04 ± 1.5 | 65.23 ± 2.0 | 78.37 ± 1.8 | 62.29 ± 2.2 | 66.50 ± 1.7 |
| | ARI | 54.90 ± 1.3 | 71.68 ± 1.7 | 83.91 ± 2.0 | 66.18 ± 1.5 | 74.55 ± 1.8 |
| | NMI | 71.71 ± 1.4 | 80.55 ± 1.9 | 87.32 ± 2.1 | 77.91 ± 2.0 | 84.35 ± 1.6 |
| GMM | Acc | 52.06 ± 1.6 | 65.77 ± 1.8 | 78.90 ± 2.0 | 61.44 ± 2.1 | 61.70 ± 1.7 |
| | ARI | 57.62 ± 1.5 | 74.76 ± 1.9 | 81.73 ± 2.1 | 64.29 ± 1.8 | 73.53 ± 1.6 |
| | NMI | 71.15 ± 1.3 | 83.75 ± 2.0 | 88.40 ± 2.2 | 76.90 ± 1.9 | 82.16 ± 1.5 |
| Spectral | Acc | 59.82 ± 1.4 | 67.92 ± 2.0 | 80.02 ± 1.7 | 60.18 ± 1.9 | 61.80 ± 1.8 |
| | ARI | 55.40 ± 1.5 | 72.18 ± 1.8 | 83.18 ± 2.1 | 60.40 ± 2.0 | 76.23 ± 1.6 |
| | NMI | 70.17 ± 1.6 | 82.95 ± 1.9 | 89.47 ± 2.2 | 75.73 ± 2.0 | 84.68 ± 1.7 |
| Agglomerative | Acc | 63.82 ± 1.7 | 70.81 ± 1.8 | 79.69 ± 2.1 | 63.94 ± 1.9 | 67.28 ± 1.6 |
| | ARI | 61.30 ± 1.4 | 75.57 ± 2.0 | 84.34 ± 2.2 | 67.95 ± 1.8 | 77.96 ± 1.7 |
| | NMI | 72.20 ± 1.5 | 85.12 ± 2.1 | 90.57 ± 2.3 | 78.55 ± 2.0 | 84.56 ± 1.8 |
| Deep_Clustering | Acc | 59.58 ± 1.6 | 67.36 ± 1.9 | 80.45 ± 2.1 | 63.18 ± 1.7 | 61.44 ± 1.8 |
| | ARI | 61.62 ± 1.5 | 74.27 ± 1.8 | 84.02 ± 2.0 | 66.50 ± 1.6 | 73.54 ± 1.9 |
| | NMI | 70.51 ± 1.4 | 83.44 ± 2.0 | 88.79 ± 2.3 | 78.11 ± 2.1 | 83.66 ± 1.7 |
| Classifier | $A_T$ | 49.19 ± 1.3 | 52.79 ± 1.8 | 66.69 ± 2.0 | 42.54 ± 1.9 | 73.23 ± 1.6 |

Table 4: Results of the ER method in the RoBERTa-base model. $A_T$ is the average accuracy by the classifier for the test of all tasks after incremental learning up to the last $T$-th task.

### NMI (Normalized Mutual Information)

NMI evaluates the mutual information between the mapped cluster labels and actual labels, normalized to account for class imbalances and size differences. Since NMI is relatively insensitive to imbalances, it is the most consistent metric for evaluating high-dimensional models and multi-class scenarios. NMI range from 0 to 1, with values closer to 1 indicating high mutual dependency between labels.

$$I(U, V) = \sum_{u \in U} \sum_{v \in V} P(u, v) \cdot \log \left( \frac{P(u, v)}{P(u) \cdot P(v)} \right) \tag{6}$$

$$H(U) = - \sum_{u \in U} P(u) \cdot \log P(u) \tag{7}$$

$$NMI(U, V) = \frac{I(U, V)}{\sqrt{H(U)H(V)}} \tag{8}$$

$I(U, V)$ is the mutual information between the cluster set $U$ and the true label set $V$. $H(U)$ is the entropy for each cluster and label. $NMI(U, V)$ is the mutual information $I(U, V)$ normalized by the geometric mean of the entropies of $U$ and $V$.

### B.2 EXPERIMENT RESULTS BY CLUSTERING ALGORITHM

We evaluated all models and IL methods with five clustering algorithms and three metrics. Most clustering algorithms recorded similar performance. For comparison with the commonly measured Average Accuracy by the classifier, we present the results of experiments with ER method on RoBERTa-base model as a representative of discriminative backbone in Table 4. Similarly, we present the results of experiments with LAMOL_g method on Pythia-410m model as a representative of generative backbone in Table 5. $A_T$ is the performance on the last task $T$ in the Average Accuracy presented in Appendix A.3.1.

In the CIL scenario, since forgetting occurs in the previous classifiers, the results measured by Clustering are better than $A_T$ in terms of NMI, ARI, and Acc. In particular, Acc outperforms $A_T$ for four datasets (Tacred, Banking77, Clinc150, and Fewrel) with more than seven incremental tasks.

| Scenario:CIL | | Task | | | | |
|---|---|---|---|---|---|---|
| Clustering Algorithm | Metric | Tacred | Banking77 | Clinc150 | Fewrel | Topic3 |
| K-means | Acc | 27.36 ± 3.2 | 67.13 ± 2.8 | 71.39 ± 4.5 | 57.23 ± 3.1 | 63.23 ± 2.4 |
| | ARI | 24.66 ± 2.6 | 71.23 ± 3.1 | 76.24 ± 4.2 | 62.12 ± 2.9 | 68.22 ± 2.7 |
| | NMI | 45.67 ± 2.3 | 81.93 ± 3.0 | 85.25 ± 3.8 | 74.33 ± 4.5 | 80.91 ± 3.1 |
| GMM | Acc | 27.77 ± 2.8 | 65.38 ± 3.4 | 71.89 ± 3.7 | 58.36 ± 4.2 | 59.27 ± 2.5 |
| | ARI | 23.97 ± 3.1 | 70.46 ± 2.6 | 77.82 ± 4.1 | 61.99 ± 3.7 | 72.23 ± 3.0 |
| | NMI | 46.03 ± 3.4 | 82.23 ± 2.9 | 84.91 ± 3.3 | 73.10 ± 4.4 | 81.85 ± 3.5 |
| Spectral | Acc | 28.74 ± 4.1 | 66.40 ± 2.5 | 71.44 ± 3.6 | 55.13 ± 3.9 | 48.80 ± 2.8 |
| | ARI | 21.46 ± 3.7 | 69.45 ± 2.7 | 76.61 ± 4.4 | 56.08 ± 4.3 | 73.90 ± 3.1 |
| | NMI | 46.38 ± 3.2 | 82.42 ± 2.8 | 86.06 ± 4.1 | 72.96 ± 3.5 | 84.48 ± 2.7 |
| Agglomerative | Acc | 28.30 ± 3.1 | 68.12 ± 2.6 | 72.16 ± 3.8 | 61.79 ± 3.7 | 76.09 ± 3.9 |
| | ARI | 24.00 ± 2.7 | 71.44 ± 3.5 | 78.11 ± 3.3 | 65.40 ± 3.1 | 80.59 ± 4.2 |
| | NMI | 46.99 ± 2.9 | 83.86 ± 2.4 | 87.65 ± 4.0 | 76.31 ± 4.1 | 83.22 ± 3.5 |
| Deep_Clustering | Acc | 28.12 ± 2.8 | 66.32 ± 3.6 | 71.23 ± 4.2 | 56.12 ± 3.5 | 65.42 ± 4.3 |
| | ARI | 24.34 ± 3.3 | 69.23 ± 3.4 | 77.24 ± 3.7 | 63.78 ± 4.1 | 71.10 ± 3.6 |
| | NMI | 46.23 ± 2.6 | 83.23 ± 3.5 | 86.11 ± 3.9 | 74.60 ± 4.2 | 82.12 ± 3.3 |
| Classifier | $A_T$ | 28.95 ± 3.2 | 51.95 ± 4.3 | 34.38 ± 3.1 | 23.09 ± 4.2 | 74.65 ± 3.4 |

Table 5: Results of the LAMOL_g method in the Pythia-410m model. $A_T$ is the average accuracy by the classifier for the test of all tasks after incremental learning up to the last $T$-th task.

On the other hand, in Topic3Dataset with only five incremental tasks, $A_T$ outperforms Acc of all clustering algorithms. This means that the more incremental tasks there are, the more forgetting in the classifier degrades the average performance, showing how biased it is to measure performance based on the classifier.

Our goal in this study is not to determine which clustering algorithm is best or which metric is the best. Therefore, we used the Spectral Algorithm, which showed average performance, for visualization and analysis in the main paper, and all the measurement results are presented in Appendix E.

## C  INCREMENTAL LEARNING METHODS

We measured BWT and FWT for representative Incremental Learning methods. Aside from a brief explanation, we adopted detailed experimental settings widely used in prior studies.

**Base** - Sequentially fine-tunes tasks. Typically, when evaluating performance through a classifier, it is known to predict only the classes of the most recently trained task for all tasks.

**ER**(Chaudhry et al., 2019) - A classical anti-forgetting technique that involves incorporating old samples. When learning a new task, a portion of old samples is included in the training process. In this study, we include one old sample per class during training.

**DER++**(Buzzega et al., 2020) - DER++ extends ER by utilizing Knowledge Distillation through the MSE loss between a teacher model and a student model, rather than simply including old samples in training. Although originally developed for the computer vision domain, we evaluated this method with both discriminative and generative backbones.

**CLSER**(Arani et al., 2022). CLS-ER builds on ER and DER++ by employing a dual-memory experience replay mechanism with fast and slow models. Like DER++, this method was initially used in the computer vision domain, but we applied it to both discriminative and generative backbones.

**L2KD**(Chuang et al., 2020) - L2KD is a method based on LAMOL, incorporating Knowledge Distillation. The teacher model learns the respective tasks first.

**LAMOL_g & LAMOL_t**(Sun et al., 2020) LAMOL is a method designed for generative models. When training on a new task, the model generates pseudo-samples of previous tasks and learns

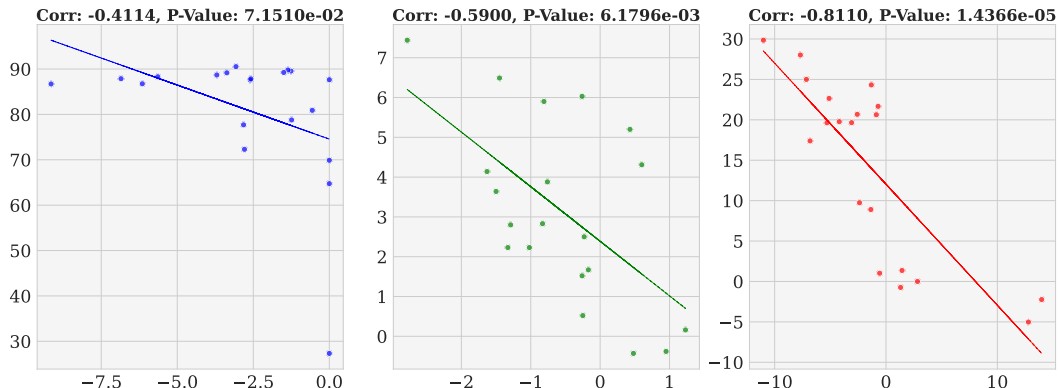

Figure 6: Pearson Correlation coefficients of three BWT, FWT measurement methods of the RoBERTa-base model in the TIL scenario. Four IL methods were applied: Base, ER, DER++, and CLSER.

them together with the new task. The key difference is that LAMOL_g does not use the gen_token, whereas LAMOL_t does.

**LAMOL_KD**(Zheng et al., 2024) - Similar to L2KD, but with the distinction that the teacher model learns all previous tasks and transfers the knowledge via Knowledge Distillation.

**PCLL**(Zhao et al., 2022) - Based on LAMOL, PCLL introduces the use of Variational AutoEncoders (VAEs) to perform Knowledge Distillation.

**SEQ***(Zheng et al., 2024) - SEQ* maximizes the backbone's anti-forgetting capability and prevents bias or forgetting in the classifier. After warming up (fine-tuning) the backbone with the first task, it is frozen, and only the classifier is newly trained for all subsequent tasks using the frozen backbone's outputs. Since the backbone remains fixed after the first task, its results do not change. While this method prevents forgetting by fixing the backbone, it also means that the backbone does not undergo the backward process of loss computation, leaving it unable to learn new knowledge or forget prior knowledge. As a result, the BWT-backbone and FWT-backbone are always 0.

**KLDA**(Momeni et al., 2025) - KLDA projects input data into a high-dimensional feature space using a kernel function and then performs Linear Discriminant Analysis (LDA) to maximize class separability. This approach effectively captures non-linear decision boundaries and is used in continual learning to enhance class distinction based on embeddings extracted from foundation models.

## D    RESULTS IN THE TIL SCENARIO

We compare the results of three measurement methods for BWT and FWT using four Incremental Learning methods across four discriminative backbones, as in the CIL scenario. As shown in Figure 6, the BWT-backbone and FWT-backbone measurement consistently recorded the highest negative correlation across all four backbones. Even in the TIL scenario, where separate classifiers are used for each task, the stability-plasticity dilemma appeared more prominently in the backbone than in the linear layer classifier.

Observing the x-axis in Figure 7, BWT values are close to 0 in the TIL scenario, as separate classifiers are used for each task. However, prior research (Zhou & Srikumar, 2021b) has suggested that classifiers can account for much of the performance, even when the model's representation is relatively weak. In comparison, BWT-backbone demonstrates a much broader range of values, revealing differences in model performance and the varying effectiveness of incremental learning methods that were obscured by classifier performance. BWT-prob, as in the CIL scenario, measures values close to 0 for all models and methods due to the use of separate classifiers.

On the y-axis of Figure 7, the FWT results for the three measurement methods show distinct differences even in the TIL scenario. FWT values exceed 60 in most cases, as they are measured on classifiers that do not know the next class, similar to the CIL scenario. Lastly, FWT-backbone re-

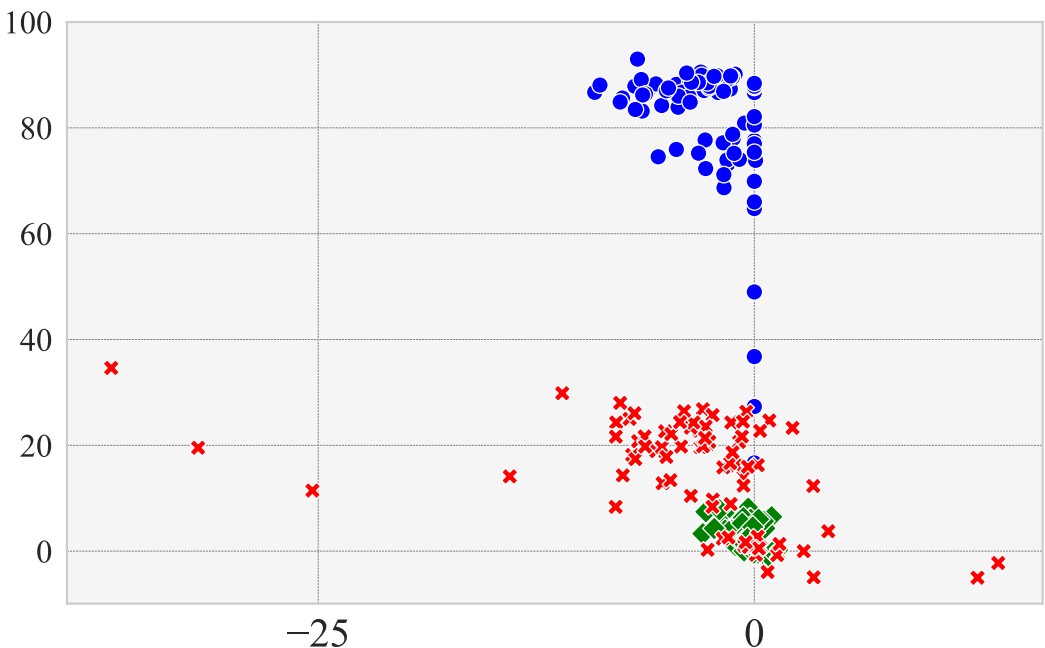

Figure 7: TIL results of experiments with four IL methods on five tasks in four discriminative backbones. The x-axis represents BWT, and the y-axis represents FWT. The evaluation metric for the clustering algorithm was the Acc of the K-means clustering algorithm.

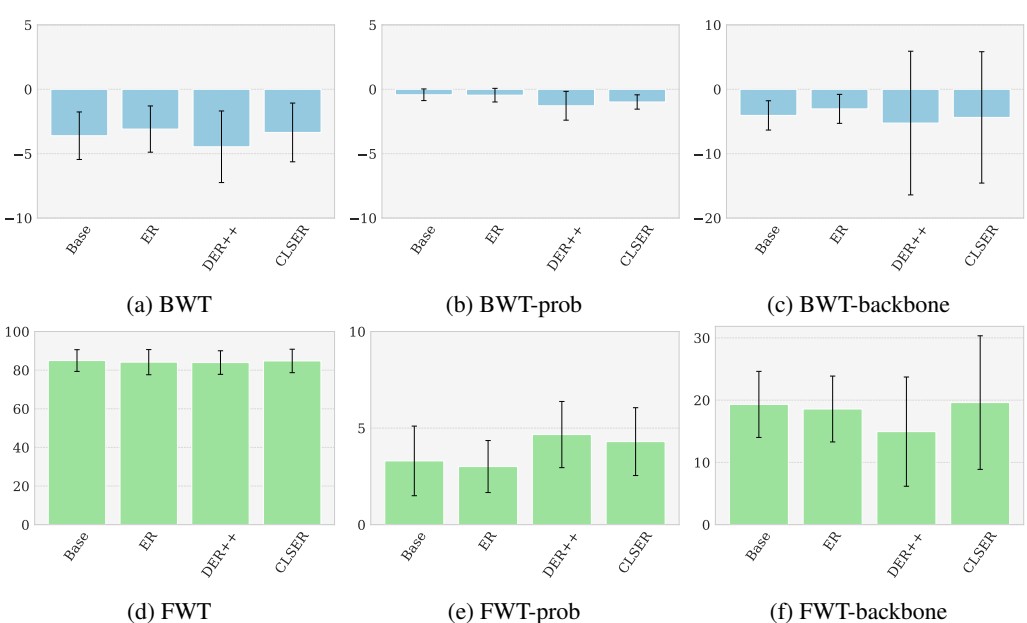

Figure 8: BWT, FWT measurement results by IL method of discriminative backbone in TIL scenario. Each Figure shows the mean and standard deviation.

veals performance improvements in the backbone as new tasks are learned, which aligns with the explanation provided for the CIL scenario in Section 4.3.

In all cases of Figures 8a, 8b, and 8c, the BWT measurement results follow the same order: ER > BASE > CLSER > DER++. (Of course, BWT-prob is trained separately on the classifier and is not learned together with the backbone.) This is because, in the TIL scenario, where classifiers

| Scenario:TIL | | Discriminative Backbone | | | |
|---|---|---|---|---|---|
| | Metric | BERT-base | BERT-large | RoBERTa-base | RoBERTa-large |
| Classifier | Acc | -0.4578 | -0.6123 | -0.5087 | -0.6342 |
| Prob Classifier | Acc | -0.5471 | -0.4982 | -0.4723 | -0.6234 |
| | Acc | -0.8421 | -0.7967 | -0.8732 | -0.8054 |
| K-means | ARI | -0.8845 | -0.8123 | -0.7987 | -0.8698 |
| | NMI | -0.8834 | -0.8291 | -0.8415 | -0.8226 |
| | Acc | -0.8023 | -0.8812 | -0.8654 | -0.7841 |
| GMM | ARI | -0.8912 | -0.8764 | -0.7998 | -0.8457 |
| | NMI | -0.8781 | -0.8534 | -0.8176 | -0.8394 |
| | Acc | -0.8107 | -0.8723 | -0.8321 | -0.7955 |
| Spectral | ARI | -0.8699 | -0.8456 | -0.8931 | -0.8744 |
| | NMI | -0.8543 | -0.8198 | -0.8411 | -0.8294 |
| | Acc | -0.8742 | -0.8543 | -0.7987 | -0.8921 |
| Agglomerative | ARI | -0.9112 | -0.8763 | -0.8194 | -0.8774 |
| | NMI | -0.8915 | -0.8381 | -0.8621 | -0.8044 |
| | Acc | -0.8142 | -0.7921 | -0.8764 | -0.8321 |
| Deep Clustering | ARI | -0.8776 | -0.8221 | -0.8442 | -0.8167 |
| | NMI | -0.8794 | -0.8472 | -0.8123 | -0.8917 |

Table 6: Pearson correlation coefficient between BWT and FWT for each metric in TIL scenario. It was measured by performing IL on five tasks in four methods (Base, ER, DER++, CLSER).

are used separately for each task, all three measurement methods are independent of classifier bias and forgetting. BWT-backbone produced results very similar to the BWT measurements in the TIL scenario, demonstrating that forgetting can be measured without the use of a classifier.

Figures 8d, 8e, and 8f compare the FWT measurement results in the TIL scenario. In Figure 8d, FWT scores all averaged above 80, showing no differences across methods. In Figure 8e, FWT-prob recorded extremely low values, averaging below 5 for all methods, with no discernible distinction. In Figure 8f, similar to Figure 3f, the results resemble those of the CIL scenario, as the backbone undergoes the same processes except for the difference in the classifiers used for CIL and TIL. The fact that FWT-backbone presents the same results in both the TIL and CIL scenarios is an impressive finding, as it consistently measures the backbone's forward learning independent of the classifier. All experimental results are presented in Table 6.

In the TIL scenario on the generative backbone, it was difficult to conduct experiments under the same conditions because some methods were optimized only for CIL. Many IL methods focus on CIL scenarios, which are much more challenging than TIL, where Base methods already perform well, using separate classifiers for each task.

## E  FULL RESULTS

We conducted the same experiment for all clustering algorithms and metrics other than those analyzed in the main paper. Looking at the results measured in the discriminative backbone in Table 1, 7, the BWT, FWT correlations measured based on Accuracy in the Classifier and Probing Classifier have little or positive correlation in all cases. The existing BWT, FWT measurement methods did not satisfy the Stability-Plasticity Dilemma at all.

On the other hand, the BWT-backbone and FWT-backbone measurement results showed a negative correlation regardless of whether Acc, ARI, or NMI was used as a metric, and the backbone showed results that conformed to the Stability-Plasticity Dilemma. The results of the generative backbone in Table 1, 7 also show the same results as the main paper. Observed Acc and Probing Acc showed a weak negative correlation between BWT and FWT, while all clustering algorithms and metrics showed a very strong negative correlation. Even considering that this result is the result of integrating

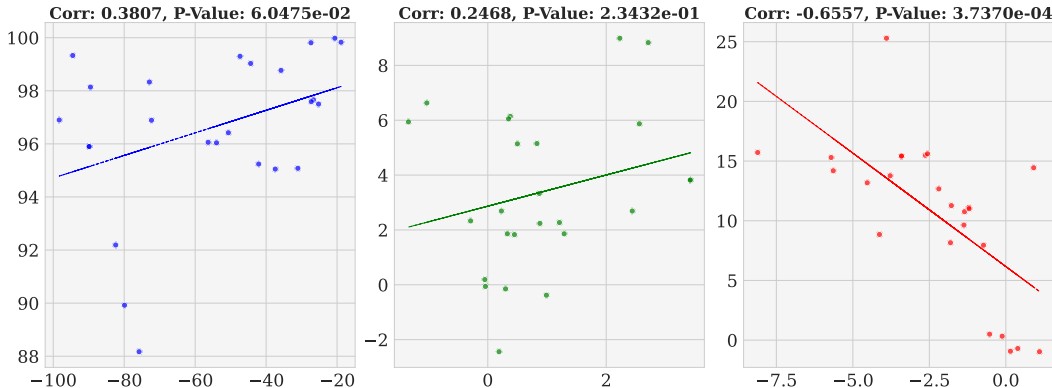

Figure 9: Pearson Correlation coefficients between BWT and FWT by three evaluation methods in the RoBERTa-base model.

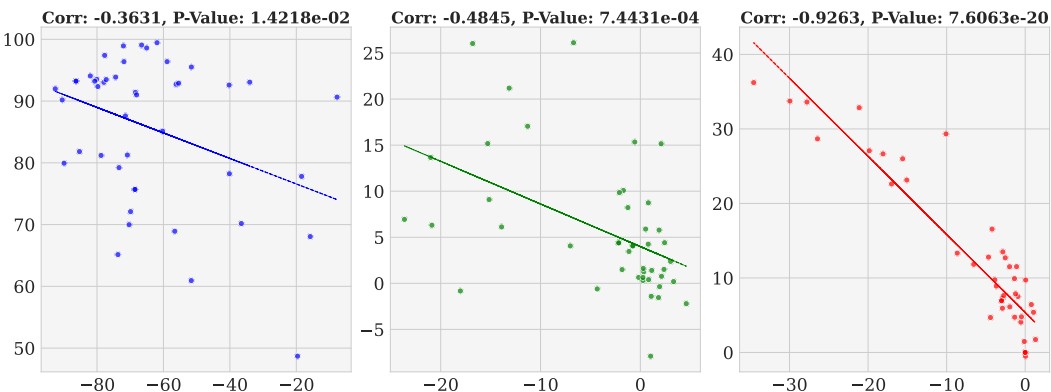

Figure 10: Pearson Correlation coefficients between BWT and FWT by three evaluation methods in the Pythia-160m model.

all IL methods and tasks, it can be seen that the trade-off relationship between Stability and Plasticity is significantly maintained during the IL process.

|  | Scenario:CIL | Discriminative Backbone | | | | Generative Backbone | | | | | |
|---|---|---|---|---|---|---|---|---|---|---|---|
|  | Metric | BERT-b | BERT-l | RoBERTa-b | RoBERTa-l | Pythia-70m | Pythia-160m | Pythia-410m | Qwen2-0.5B | Qwen2.5-0.5B | Qwen3-0.6B |
| Classifier | Acc | 0.4225 | 0.1327 | 0.3752 | 0.2385 | -0.5142 | -0.3729 | -0.3654 | -0.5300 | -0.3800 | -0.3670 |
| Prob Classifier | Acc | 0.6402 | 0.1063 | 0.2417 | 0.1046 | -0.4239 | -0.4725 | -0.6339 | -0.4350 | -0.4870 | -0.6280 |
| K-means | Acc | -0.6246 | -0.5587 | -0.4873 | -0.5182 | -0.8368 | -0.8660 | -0.8481 | -0.8420 | -0.8580 | -0.8550 |
|  | ARI | -0.6612 | -0.5196 | -0.5275 | -0.5441 | -0.8125 | -0.8001 | -0.8257 | -0.8150 | -0.8080 | -0.8280 |
|  | NMI | -0.6459 | -0.6073 | -0.5792 | -0.6741 | -0.8204 | -0.9212 | -0.8874 | -0.9120 | -0.8930 |
| GMM | Acc | -0.5144 | -0.5751 | -0.4901 | -0.4323 | -0.7623 | -0.7439 | -0.7801 | -0.7700 | -0.7480 | -0.7820 |
|  | ARI | -0.6693 | -0.6854 | -0.6191 | -0.5172 | -0.8152 | -0.8071 | -0.8537 | -0.8180 | -0.8090 | -0.8570 |
|  | NMI | -0.5547 | -0.5923 | -0.6147 | -0.6085 | -0.7862 | -0.8374 | -0.8376 | -0.7900 | -0.8340 | -0.8390 |
| Spectral | Acc | -0.6883 | -0.6145 | -0.6112 | -0.6031 | -0.8169 | -0.8257 | -0.8602 | -0.8200 | -0.8280 | -0.8580 |
|  | ARI | -0.6802 | -0.6263 | -0.6811 | -0.6478 | -0.8173 | -0.8523 | -0.8064 | -0.8200 | -0.8540 | -0.8080 |
|  | NMI | -0.6184 | -0.6391 | -0.6519 | -0.6102 | -0.8185 | -0.8841 | -0.8762 | -0.8200 | -0.8820 | -0.8730 |
| Agglomerative | Acc | -0.4722 | -0.5068 | -0.6123 | -0.4597 | -0.7824 | -0.7369 | -0.7046 | -0.7880 | -0.7380 | -0.7070 |
|  | ARI | -0.7439 | -0.6874 | -0.6012 | -0.7325 | -0.8647 | -0.8099 | -0.8865 | -0.8660 | -0.8130 | -0.8890 |
|  | NMI | -0.8962 | -0.7145 | -0.7298 | -0.6242 | -0.8751 | -0.9198 | -0.8772 | -0.9170 | -0.8760 |
| Deep Clustering | Acc | -0.4978 | -0.4936 | -0.6142 | -0.5827 | -0.8034 | -0.8297 | -0.8891 | -0.8100 | -0.8330 | -0.8870 |
|  | ARI | -0.4781 | -0.5279 | -0.5831 | -0.4852 | -0.8457 | -0.8192 | -0.7931 | -0.8480 | -0.8200 | -0.7980 |
|  | NMI | -0.5221 | -0.5394 | -0.5984 | -0.6273 | -0.8702 | -0.7244 | -0.7916 | -0.8750 | -0.7220 | -0.7890 |

Table 7: Spearman correlation coefficient between BWT and FWT for each metric in CIL scenario.

### E.1 MODEL SIZE & CAPACITY

In Figure 11, we measured BWT in backbone, FWT in backbone, BWT in backbone + FWT in backbone using the base method in three groups based on representative LLMs (GPT(Pythia), Llama, and

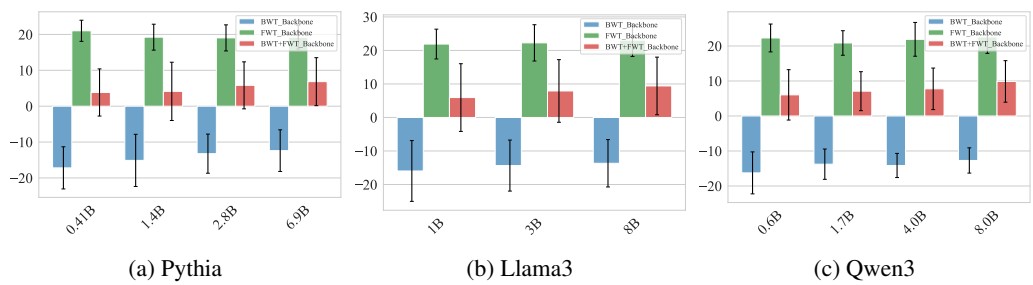

(a) Pythia   (b) Llama3   (c) Qwen3

Figure 11: BWT-Backbone, FWT-Backbone, BWT-Backbone + FWT-Backbone results in Pythia, Llama, Qwen, measured in five benchmarks, only Base method.

| Hyperparameter | Default |
|---|---|
| lm_head_lr | e.g. 5e-4 |
| Optimizer | AdamW |
| Loss (LM) | Cross-entropy |
| training_epoch | 3 |
| Batching | Same as backbone training loop |

Table 8: Hyperparameters for LM-head fine-tuning.

Qwen). If we consider BWT+FWT-backbone as the capacity of the model, the capacity increases as the size increases in the same model series. In particular, as the model size increases, BWT-backbone decreases, indicating a positive relationship between the model size and anti-forgetting ability. On the other hand, the FWT-backbone were nearly constant across all models and sizes. Unlike the IL methods that showed no difference, we found that within the same family of models, FWT remained constant while BWT improved as the size increased. The constant FWT even with larger model sizes should be considered when designing future IL methods.

## F   LM-HEAD FINE-TUNING DETAILS

**Setup.** During training, if LM_HEAD_FINETUNE is enabled, the backbone and the LM head are updated jointly at each step. During evaluation, we freeze the backbone and perform a short *mini-finetuning* of the LM head on the training split of the current task (see Section F.1).

**Label tokenization.** For each instance, we tokenize the gold textual label and take the first token id as target:

$$y^{\text{tok}} = \text{Tokenizer}(\text{label})[0].$$

Given hidden feature $\mathbf{h} \in \mathbb{R}^d$ from the backbone and output embedding $W \in \mathbb{R}^{V \times d}$, the vocabulary logits are

$$\mathbf{z} = W\,\mathbf{h} \in \mathbb{R}^V.$$

**Base LM-head loss.**

$$\mathcal{L}_{\text{LM}} = \text{CE}\big(\mathbf{z}, y^{\text{tok}}\big).$$

**Optimization.** The LM head parameters are optimized by AdamW with learning rate $\eta$ (default chosen as 5e$-$4 in our runs). Backbone and optional external classifiers are trained with a separate optimizer.

### F.1   EVALUATION-TIME LM-HEAD FINETUNE

Before evaluation on task $t$, the backbone is frozen and the LM head is adapted for $E$ epochs (typically 1–3) on the training split of task $t$ using $\mathcal{L}_{\text{LM}}$. After evaluation, LM-head weights are restored to their original state to avoid leakage across tasks.

