# OpenReview forum: "Transfer is All You Need: Revisiting the Stability–Plasticity Dilemma through Backward and Forward Transfer in PLMs"
_ICLR.cc/2026/Conference — Submitted to ICLR 2026_

### Official Review · Reviewer_X4ao · 2025-10-16

**Soundness:** 2
**Presentation:** 3
**Contribution:** 2
**Rating:** 2
**Confidence:** 5

**Summary:**

This paper applies the evaluation metrics of Backward Transfer (BWT) and Forward Transfer (FWT) directly to the backbone, avoiding the drawbacks of traditional computation methods. The experiments verify the differences between the two evaluation approaches and propose Just LM-Head Tuning (JLT), which performs lightweight training only on the classifier to adapt to incremental tasks. Although the experiments demonstrate the effectiveness of the JLT method, they do not address the core issue of the paper — the stability–plasticity dilemma.

**Strengths:**

1. The paper verifies that forgetting typically occurs in the classifier rather than in the backbone.
2. The paper proposes evaluating BWT and FWT on the backbone to avoid the bias introduced by the classifier.
3. The paper is generally clearly written.

**Weaknesses:**

1. Although the newly defined FWT and BWT metrics partially avoid the drawbacks of traditional computation methods, they also introduce new instabilities (as discussed in the Question section).
2. The new evaluation approach significantly increases the experimental cost and duration, making rapid validation difficult and limiting its applicability to other related research areas.
3. Regarding the stability–plasticity dilemma raised in the Introduction, the JLT method itself does not resolve this issue; instead, it proposes a more efficient way to utilize the backbone, rather than addressing the inherent forgetting problem that may occur in the backbone or the head during incremental learning.
4. The paper’s underlying assumption is that “the backbone is more stable, and most forgetting occurs in the classifier.” However, the subsequent experiments mainly reinforce this existing conclusion. As discussed in related works [1] and [2], both have addressed similar points, and the proposed evaluation method and JLT approach seem to follow naturally from this pre-existing assumption.

[1] Probing Representation Forgetting in Supervised and Unsupervised Continual Learning, CVPR 2022.

[2] Learn or Recall? Revisiting Incremental Learning with Pre-trained Language Models, ACL 2024.

**Questions:**

About FWT:

(1) When evaluating FWT, does the use of a temporary classifier introduce new bias? Training a classifier for each new task $i$ means that the FWT score will heavily depend on the training quality of this temporary classifier. If the classifier is well or poorly trained, it may overestimate or underestimate the actual transferability of the backbone.

(2) Since the classifier used for evaluation is temporary and unrelated to the final classifier used in the incremental model, could this make the evaluation inaccurate with respect to the final transfer performance we truly care about? Although traditional methods using an external classifier are detached from the intermediate process, the classifier is at least fixed and well-optimized. In contrast, the proposed method is process-dependent and introduces new variables related to training dynamics and hyperparameter sensitivity.

About BWT:

(1) Using clustering as a proxy task to evaluate performance on supervised learning tasks raises the question: can good clustering performance truly be considered equivalent to good classification performance?

(2) The paper experiments with multiple clustering algorithms, but the results appear to vary. Does this indicate that the newly defined BWT is not a stable metric and is sensitive to the choice of clustering algorithm, thereby reducing its comparability?

(3) Clustering algorithms inherently lose some of the fine-grained information that is crucial in tasks, a limitation that the paper does not seem to discuss.

**Details Of Ethics Concerns:**

The supplementary code package includes a README file that reveals the authors’ identities. This violates the double-blind review policy. I recommend the editorial team review this issue before proceeding with the evaluation.

---

> ### Author Response · Authors · 2025-11-24
> **Response to the Concerns [1/3]**
>
> First and foremost, we were truly honored and deeply appreciative of your insightful and thorough review. We sincerely thank you for the considerable attention and thoughtful engagement you have given to our work. However, we recognize that our explanation may have been insufficient, resulting in certain aspects being conveyed in a manner different from our intention. We would therefore like to provide our responses as follows.
>
> **Weakness 2**
>
> We agree with the concern you raised. As you mentioned, introducing an additional evaluation process may increase the experimental cost and duration. However, our intention is not to argue that this procedure should be incorporated into all future experimental frameworks. Rather, by demonstrating that the performance measured at the backbone level can differ substantially from that measured at the classifier level, we hoped to highlight an observation that may inform future research directions in IL methods.
>
> Moreover, since the evaluation only requires applying basic clustering algorithms to the representations of an already trained backbone, the computational cost is significantly lower than the evaluation method used in prior work [1], which relies on training a separate probing classifier. Below are the computational overhead measurements for Pythia-410M:
>
> | Method | Topic3 | FewRel | CLINC150 | TACRED | Banking77 |
> | --- | --- | --- | --- | --- | --- |
> | Probing Classifier | 23m 15s | 4m 28s | 1m 32s | 1m 7s | 54s |
> | K-means | 9m 16s | 1m 22s | 15s | 4s | 7s |
> | GMM | 8m 19s | 1m 18s | 12s | 4s | 8s |
> | Spectral | 7m 46s | 1m 14s | 19s | 4s | 9s |
> | Agglomerative | 13m 20s | 1m 50s | 4s | 2s | 2s |
> | Deep Clustering | 18m 19s | 2m 13s | 26s | 9s | 15s |
>
> In practice, during our research, we found that training the classifier used in prior evaluations required substantial computational effort, making it difficult to evaluate across many models and IL methods. For this reason, we were genuinely impressed that you identified this point. Thank you once again for your valuable and insightful feedback.
>
> [1] Learn or Recall? Revisiting Incremental Learning with Pre-trained Language Models, ACL, 2024.
>
>
> **Weakness 3**
>
> Thank you for sharply identifying the areas where our claims and reasoning were unclear. We fully acknowledge this shortcoming. However, we would like to reiterate that the objective of our work was *not* to address the stability–plasticity dilemma. The intended logical flow of our argument is as follows:
>
> 1. Based on classifier-based evaluation, IL methods have been presented as if they successfully resolved catastrophic forgetting.
> 2. However, this largely overlooked FWT, and when evaluating performance at the backbone level using our proposed FWT metric, we observed that BWT and FWT exhibited a trade-off relationship.
> 3. Even under the base learning approach, the backbone was neither forgetting everything nor learning entirely from scratch. It retained a sufficient degree of anti-forgetting and demonstrated forward transfer from its existing performance.
> 4. Therefore, effectively leveraging only the backbone can already provide sufficient backward and forward transfer.
>
> We recognize that mentioning the stability–plasticity dilemma in the title and in parts of the paper may have made the purpose of our study difficult to interpret and potentially confusing. In the revised version, we will improve the structure so that the paper clearly follows the logical flow described above. Once again, we sincerely thank you for your deep insight.

---

> ### Author Response · Authors · 2025-11-24
> **Response to the Concerns [2/3]**
>
> **Weakness 4**
>
> This is indeed an excellent observation. However, we would like to clarify the differences between the prior studies you referenced and our work as follows.
>
> [1] conducted their analysis in the task-incremental learning (TIL) setting. As described in Section 2.1 “Problem Setup”, TIL differs from class-incremental learning (CIL) in that the classes do not overlap across tasks. TIL has already been largely addressed in the current literature, and we also provide separate experimental results for this setting in Appendix D “Results in the TIL Scenario.” In the Limitations and Future Work section of [1], the authors explicitly note that their study is restricted to TIL, and that CIL poses additional challenges, which they leave as future work.
>
> [2], which is also one of our main baselines, does not address forward transfer at all, due to the inherent difficulty of defining FWT as described in our paper. By proposing a new evaluation procedure and FWT metric, we aimed to shed light on this unexplored aspect. There is also a significant difference in terms of BWT. This prior work performs probing using a separate linear layer that does not participate in the IL process, thereby presenting results that make it appear as if catastrophic forgetting hardly occurs even in the base method (see Figure 3(c) of that work).
>
> In contrast, Figures 3 and 5 of our study clearly indicate that varying degrees of forgetting occur depending on the IL method. Under our evaluation approach, catastrophic forgetting was indeed observed in the base method, and techniques such as Replay, Knowledge Distillation, and Variational Autoencoders demonstrated stronger anti-forgetting performance compared to the base approach.
>
> Therefore, we believe that our work provides much more detailed and deeper empirical observations than prior studies. Thank you once again.
>
> [1] Probing Representation Forgetting in Supervised and Unsupervised Continual Learning, CVPR 2022.
>
> [2] Learn or Recall? Revisiting Incremental Learning with Pre-trained Language Models, ACL 2024.
>
> **Question**
>
> **FWT**
>
> 1. I am truly impressed by the insight with which you identified this point. We fully agree, and the introduction of a new classifier can indeed cause substantial issues depending on training quality. However, we would like to clarify that the concern you raised actually arises from the evaluation method described in Section 4.2, “BWT & FWT in Probing Classifier,” as well as the prior work [1] that followed this approach—whereas our proposed JLT is fundamentally different. As you pointed out, introducing a new classifier for each task may lead to significant bias or variability. In contrast, JLT does not introduce a new classifier; instead, it retrains the existing LM head of the current model. Furthermore, to support the possibility of bias or variance during LM head training, we additionally measured and reported the backbone’s BWT and FWT. As a result, rather than producing entirely different BWT and FWT values across the three evaluation methods (as seen in Figures 3 and 5), we were able to present results in Table 2 that closely align with the measurements obtained from the backbone.
> 2. We believe this misunderstanding also stems from insufficient clarity in our explanation, and we sincerely ask for your understanding. Please refer to the following details:
>     1. The method in prior work [1] uses a completely separate classifier and involves various procedures—such as pre-allocation and warm-up—to optimize this temporary classifier.
>     2. In contrast, our approach uses the LM head that is inherently part of the model, pre-trained together with the backbone, and continuously involved throughout the IL process.
>     3. The hyperparameter settings for the LM head are designed for retraining the LM head—after completing the IL process—using the backbone’s features, in order to transfer the backbone’s capability.
>     4. In prior work [1], the backbone is frozen after task 1, preventing it from acquiring additional knowledge, and a completely new classifier—separate from the original model—is trained.
>
> We believe that the points you raised regarding FWT resulted from a lack of clarity in our explanation. In the revised version of the paper, we will include a detailed description of JLT as well as a comparison table with other methods. We kindly invite you to refer to the updated version once uploaded.
>
> [1] Learn or Recall? Revisiting Incremental Learning with Pre-trained Language Models, ACL, 2024.

---

> ### Author Response · Authors · 2025-11-24
> **Response to the Concerns [3/3]**
>
> **Questions**
>
> **BWT**
> **1 & 3.** We fully agree with the concerns you raised. As you pointed out, evaluation through clustering cannot be perfectly equivalent. In response to this, we referred to prior studies that attempted to evaluate models using their representations [1, 2, 3]. We also emphasized multiple times that this is **not** intended to serve as a complete evaluation method, but rather as an *“Auxiliary Evaluation,”* and we clearly stated this in the limitations section. In addition, instead of relying solely on clustering accuracy, we incorporated ARI and NMI—metrics that compensate for the fine-grained information loss that may occur due to clustering algorithms, as you mentioned. We were fully aware of the limitations you pointed out and attempted to address them by (1) presenting the method as an Auxiliary Evaluation, (2) introducing ARI and NMI metrics, and (3) explicitly acknowledging the limitations in the paper. Once again, we were deeply impressed by your insight in identifying points that closely align with challenges we also experienced throughout our research.
>
> **2.** Our intention in presenting a wide range of experiments was to demonstrate the robustness of the measured results across various settings. The key point is not the variation introduced by different clustering algorithms, but rather the substantial discrepancy between performance measured at the classifier level and that measured at the backbone level. Moreover, BWT is a highly general metric that can be computed using the classifier, the probing classifier, and clustering-based evaluation on the backbone. It is a metric derived from processing performance scores evaluated across tasks.
>
> Thank you for sharing such thoughtful and in-depth comments regarding BWT. We were surprised that you highlighted issues that closely reflected our own observations during the research process, and we appreciate your understanding that we have already given considerable thought to these concerns and taken multiple steps to address them.
>
> [1] Continual learning for sentence representations using conceptors. NAACL, 2019
>
> [2]  A closer look at how fine-tuning changes bert. ACL, 2021
>
> [3] Directprobe: Studying representations without classifiers. NAACL, 2021
>
> We sincerely appreciate your review, and it has been both a pleasure and an honor to have this discussion with you. Please do not hesitate to let us know if you have any further questions.
>
> Thank you.
> The Authors

---

> > ### Comment · Reviewer_X4ao · 2025-11-26
> >
> > I thank the authors for their responses to my questions. Regarding W3, I believe the authors have addressed my concerns. I encourage the authors to incorporate the key clarifications into the revised submission. For now, I have raised the score.

---

### Official Review · Reviewer_JJQR · 2025-10-28

**Soundness:** 2
**Presentation:** 2
**Contribution:** 2
**Rating:** 2
**Confidence:** 4

**Summary:**

The authors conduct an in-depth evaluation of the stability–plasticity dilemma in Pretrained Language Models (PLMs). They argue that most of the existing literature focuses on accuracy, while little attention has been paid to metrics such as backward and forward transfer. As noted by the authors, this is partly due to some “controversial” aspects related to the classifier and its handling. In this regard, they evaluate three different strategies for assessing these metrics and apply them to various models. According to their results, forgetting issues appear to be more related to the classifier than to the backbone, which, in turn, seems to exhibit an anti-forgetting effect, an observation consistent with previous studies. Finally, the authors propose a strategy called Just LM-Head, which involves training the classifier only after the incremental learning phase. This approach appears to achieve state-of-the-art results on several language tasks.

**Strengths:**

The intention of this paper is noteworthy, as it attempts to move beyond standard evaluation protocols to better understand which components of the model are responsible for forgetting issues.
The evaluation is, in principle, extensive, covering several incremental learning approaches, paradigms, and tasks. Moreover, it focuses on recent language models, offering a renewed and fresh perspective on a longstanding problem.

**Weaknesses:**

**Justification of Results**
- Just LM-Head Tuning (JLT): the authors propose to train a new classification head after the incremental learning phase. According to Table 2, this leads to remarkably high gains, with improvements of around +40–45% in accuracy. However, it is unclear how such a result is possible. During the incremental learning phase, the model should have already learned the tasks; therefore, it is not evident how retraining a new classifier after this phase could solve all the forgetting-related issues. I find no empirical, theoretical, or intuitive justification to understand or trust this result. Moreover, the preceding sections do not adequately introduce the rationale behind this approach.

**Clarity**
- Just LM-Head Tuning (JLT): In the abstract and introduction, it is difficult to clearly get the difference between the proposed approach and existing methods that freeze the backbone and train only the final classification layer. I suggest revising these sections to better highlight the distinction and possibly reconsidering the name of the method, as the current one may be somewhat confusing.
- Section 4.2: This section is unclear. The authors mention that there is a separate classifier that does not participate in the incremental learning process, yet it is later used for evaluation. It is not clear how the weights of this classifier are obtained if it does not take part in the incremental learning phase. Consequently, since the procedure for computing this separate classifier is not well explained, it becomes difficult to interpret the results derived from this analysis.
- Section 5.2: In the bullet list, the first two points appear to be in contradiction. Have I misunderstood them, or is there an inconsistency that should be clarified?

**Methodology**
- Section 4.3.1: the authors propose an approach based on clustering to compute the classification rule. But, the clustering step could introduce an additional failure point in this process. Hence, why do the authors do not rely on the simplest KNN classifier? This is much more standard in literature.

**Organization and Presentation**
- The organization of the paper is weak. Although the authors report many tests and results, it is often unclear which experiments are truly significant, reliable, or worth emphasizing. As a result, the presentation feels flat, and the reader struggles to identify the key takeaways. The most important findings should be highlighted more effectively—possibly through clearer structuring, dedicated summary boxes, or explicit remarks within the text.

**Overclaims**
-Lines 051–052: “existing IL methods have focused only on overcoming catastrophic forgetting, and mainly evaluate the average accuracy for each task.” This statement is not accurate. Numerous studies have already addressed these specific aspects, making the authors’ claim reductive with respect to the existing literature. For instance, in [1], the authors explicitly investigate the preparation for future tasks; similarly, many other works consider continual learning from both perspective, i.e., mitigating forgetting of past tasks and facilitating learning of future ones.

- Personal comment: I personally find the idea that there could be an absolute answer to the question “Is forgetting mainly caused by the classifier or by the backbone?” quite questionable. In my view, this strongly depends on the experimental setting. Based on my experience, the more the downstream tasks diverge from the pre-trained model’s domain, the more the backbone needs to adapt — and consequently, the more backbone-related forgetting issues become relevant. Therefore, I believe this is a highly complex question that is difficult to address with full rigor.

[1] Boschini, M., Bonicelli, L., Buzzega, P., Porrello, A., & Calderara, S. (2022). Class-incremental continual learning into the extended der-verse. IEEE transactions on pattern analysis and machine intelligence, 45(5), 5497-5512.

**Questions:**

See section above.

---

> ### Author Response · Authors · 2025-11-24
> **Response to the Concerns [1/3]**
>
> Thank you very much for taking the time to examine our work in such depth. We were truly impressed by your keen insight and thoughtful review, and it is a great honor for us to have this discussion with you. Below, we provide our responses to the points you raised. Once again, we kindly ask for your continued generous review.
>
> **Justification of Results**
> First, we would like to apologize for the lack of clarity in our earlier explanation regarding this point. However, this does not involve training a new classification head separately. The new classifier-learning–type approaches you mentioned have been explored in other studies as well and are similar to the Probing Classifier described in Section 4.2 of our work. The JLT we propose refers to the LM head of the model that continues to ‘participate throughout the IL process’. To justify the validity of our experimental results, we provide the following explanation of the presentation and logical flow of our experiments.
>
> 1. We presented the BWT and FWT measured using the three evaluation methods described in Figure 1 and Section 4, as shown in Figures 3 and 5.
>
> 2. The results indicated that catastrophic forgetting was severe in the classifier (LM head), whereas the backbone exhibited sufficient anti-forgetting capability. This phenomenon has been partially discussed in prior studies [1]. However, because those studies reported results using a separate classifier, they depicted the base methods as if catastrophic forgetting scarcely occurred.
>
> 3. We focused on the fact that the three evaluation methods produced entirely different results, and based on the observation that the performance of the backbone is not fully transferred to the LM head (classifier), we proposed JLT.
>
> 4. Given the capability retained by the backbone, the performance of JLT is not particularly surprising. The issue was that catastrophic forgetting was occurring due to classifier bias, despite the backbone maintaining sufficient performance.
>
>
> **Clarity**
>
> 1.Thank you sincerely for your suggestion. As you noted, another reviewer also misunderstood JLT, and our explanation was insufficient. In the revised version of the paper, we plan to present a separate method figure for JLT and clarify the explanation.
>
> The key difference from existing approaches lies in ‘whether the backbone is updated during the IL process’. In prior studies [1], [2], in order to prevent catastrophic forgetting, only the first task (or a subset of tasks) is trained, after which the backbone is frozen throughout the IL process and only the LM head (classifier) is trained. Because this preserves performance on previous tasks, it is highly effective in terms of the average accuracy metric; however, in reality, the backbone is unable to acquire new knowledge.
>
> In contrast, the JLT we propose involves both the backbone and the LM head participating throughout the entire IL process. Since the backbone maintains its performance while the LM head develops a bias toward strongly remembering only the most recently learned task, retraining the LM head just once using the backbone’s representation is sufficient. Therefore, JLT (1) allows the backbone to continue acquiring knowledge, and (2) does not require any additional structures such as a separate classifier.
>
> 2.We will add a precise definition of this in the Appendix section. Details regarding the evaluation method are provided in Figure 1. This evaluation procedure involves training a separate classifier using the backbone at each stage after learning each task. The hyper-parameter settings for training the probing classifier are presented in Appendix A.2 (Implementation Details).
>
> 3.The two points you raised do not conflict with each other, and both statements are correct.
>
> (1) As shown in Figures 3(c) and 4(c), unlike the classifiers in Figures 3(a) and 4(a), the backbone retained a considerable level of anti-forgetting capability.
>
> (2) As can be observed by comparing Figures 3(c) and (f) and Figures 4(c) and (f), the BWT and FWT of the backbone exhibited a trade-off relationship, and the corresponding correlation coefficients are presented in Table 1. These two observations are not contradictory. We clarified this because prior work [1] described the base method as if catastrophic forgetting hardly occurred (see Figure 3(c) of that work), whereas our experimental results differed. Catastrophic forgetting occurred to some extent across IL methods, including the base method, and there existed a negative correlation between backward transfer and forward transfer.
>
> [1] Learn or Recall? Revisiting Incremental Learning with Pre-trained Language Models, ACL, 2024
>
> [2] Continual learning using a kernel-based method over foundation models, AAAI, 2025

---

> ### Author Response · Authors · 2025-11-24
> **Response to the Concerns [2/3]**
>
> **Methodology**
>
> We appreciate the suggestion and agree that kNN is a widely used tool for evaluating learned representations. However, our objective in this work is slightly different: we aim to assess the intrinsic, label-agnostic structure of the backbone features, rather than the performance of yet another classifier defined on top of them.
>
> First, kNN is still a supervised decision rule that relies on a labeled reference set. Its performance depends not only on the geometry of the feature space, but also on how this reference set is constructed (e.g., which exemplars are stored, how many samples per class, whether we use all training data or a memory-limited subset, etc.). In incremental learning scenarios, these design choices are particularly consequential and would entangle our evaluation with memory and rehearsal configurations. In contrast, our clustering-based evaluation groups features without labels, and uses labels only post hoc to compute ARI/NMI. This provides a more head-agnostic and memory-agnostic view of the representation itself.
>
> Second, kNN introduces additional hyperparameters and design decisions, most notably the choice of kkk. There is no canonical value for kkk, and the resulting accuracy can vary substantially across different settings, making it less clear which configuration should be treated as the “ground truth” evaluator. In our clustering setup, by contrast, the number of clusters KKK is naturally determined by the number of ground-truth classes. Thus, KKK is fixed by the problem definition rather than treated as a tunable hyperparameter, yielding a simple, reproducible evaluation protocol.
>
> Third, kNN focuses on very local neighborhoods and is highly sensitive to local density, class imbalance, and noisy neighbors. Our goal, however, is to capture the global class structure in the feature space: how well samples from the same class form coherent groups and how clearly they are separated from other classes. Clustering combined with ARI/NMI directly evaluates this global partition, which aligns more closely with the representation-level properties we aim to study.
>
> Moreover, in typical incremental learning setups, both encoder-based and decoder-based architectures ultimately rely on a specific token-level interface for classification. For encoder models (e.g., BERT-style), performance is usually evaluated by training a classifier head on top of the [CLS] representation. For decoder-only models, classification is often implemented through label tokens in the LM head, where the model compares the logits of different label tokens given the input. In both cases, successful classification presupposes that the backbone features form coherent, well-separated regions for each class: samples from the same class should map to similar representations, while samples from different classes should be pushed apart. Our clustering-based evaluation directly measures this underlying geometric property of the backbone representation, independently of any particular head or label-token parameterization. Instead of training another classifier or label-token head and evaluating its performance, we ask whether the learned features themselves already exhibit the class-specific grouping structure that such token-level classifiers rely on.
>
> For these reasons, we believe clustering-based evaluation is more suitable for our specific objective of analyzing the structure and quality of backbone representations in incremental learning. That said, kNN-based evaluation is complementary, and incorporating it as an additional analysis is an interesting direction for future work, rather than a replacement for the clustering-based protocol we propose.

---

> ### Author Response · Authors · 2025-11-24
> **Response to the Concerns [3/3]**
>
> **Organization and Presentation**
>
> We sincerely apologize for making the paper difficult to understand. As you pointed out, we acknowledge that we presented too many experiments without sufficient emphasis, and we will significantly improve this in the revised version. For example, we plan to move the discussion on model capacity to the appendix and enrich the structure that connects the evaluation method to JLT. Below is the intended logical flow we aimed to present in order to highlight the novelty of our work:
>
> 1. Based on classifier-based evaluation, IL methods have been presented as if they successfully addressed catastrophic forgetting.
> 2. However, this largely overlooked FWT, and by measuring performance at the backbone level using our proposed FWT metric, we found that BWT and FWT exhibited a trade-off relationship.
> 3. In contrast to the measurements obtained from the conventional classifier, even under the base learning setup, the backbone was neither forgetting everything nor learning entirely from scratch. It retained a sufficient level of anti-forgetting and demonstrated forward transfer from its already existing performance.
> 4. Therefore, even leveraging only the backbone can achieve sufficient backward and forward transfer, and it becomes crucial—across all types of IL methods—to transfer the backbone’s capability to the LM head after full model training.
>
> **Overclaims**
>
> We humbly accept your comment, and we will adopt a more measured tone in the revised version of the paper. However, even in the study you referenced, FWT was reported, yet forward transfer itself was not demonstrated. As we stated in Section 2.2 — “The biggest reason is that in order to measure how well a new task is learned, ‘a_{i−1,i}’ and ‘a_{i,i}’ must be compared. But before learning the n-th task, the classifier cannot predict the n-th task at all.” — the existing definition of the FWT metric made it fundamentally difficult to measure forward transfer. Addressing this limitation was the main motivation of our work.
>
> Your personal comment was very thought-provoking, and we fully agree. This can indeed be highly sensitive to factors such as the domain of the downstream task, the choice of model backbone, and parameter configurations. However, prior studies have also reported that catastrophic forgetting is more severe in the classifier than in the backbone [1]. Our extensive experimental results further support this claim with greater detail. We believe the trends are sufficiently demonstrated by experiments conducted on eight datasets, three types of backbones (GPT, Llama, and Qwen), multiple model sizes, and multiple IL methods. For example, in Figures 3 and 4, despite using the same trained backbone, entirely different patterns emerged depending on the evaluation method. That said, as you pointed out, issues related to the training process—such as how training loss should be stabilized—certainly remain valid and unresolved, and we agree with this point.
>
>
> Thanks to your deep insight, we were able to reflect on various perspectives with both joy and appreciation. It is unfortunate that our explanations and clarity were insufficient overall. However, we believe that many of these issues can be improved through clearer descriptions and restructuring, and we are fully incorporating these revisions into the updated paper. Please know that we remain open to all further discussion. Once again, we sincerely thank you for your thoughtful and insightful review.

---

> > ### Comment · Reviewer_JJQR · 2025-11-26
> >
> > Thank you to the authors for their thoughtful and open-minded response. I genuinely appreciated the tone of the rebuttal, it was a pleasure to read. For this reason, I decided to increase my score, although it is still not positive.
> >
> > That said, I believe the paper would benefit from further improvements.
> > - Even after carefully reading the rebuttal, I still do not clearly understand what JTL does methodologically. I would not be able to explain the core idea to a colleague or a student, which suggests that the exposition could be made substantially clearer.
> > - I remain unconvinced by the discussion around k-NN. While it is true that k-NN has several limitations, I am not fully persuaded that the proposed solution is significantly less affected by related issues.

---

> > > ### Author Response · Authors · 2025-11-28
> > > **Thank you for your thoughtful feedback. Please find our additional response. [1/2]**
> > >
> > > Thank you sincerely for viewing our intentions in a positive light. We are deeply grateful for your thoughtful understanding. Below, we provide detailed and concrete explanations for the two unresolved points you highlighted.
> > >
> > > **Explanation of JLT**
> > >
> > > Intuitively, JLT works as follows:
> > > because there is a large mismatch between the performance measured at the backbone level and the performance measured at the LM head, JLT enables the LM head to inherit the performance of the backbone.
> > >
> > > For example, Qwen3.0-0.6B model on the TACRED dataset, the measurement results are shown below.
> > >
> > > **Backbone (Spectral Algorithm NMI)**
> > > | Task1   | Task2   | Task3   | Task4   | Task5   | Task6   | Task7   | Task8   |
> > > | ------- | ------- | ------- | ------- | ------- | ------- | ------- | ------- |
> > > | 76.294  | 68.2249 | 62.995  | 49.5796 | 46.2606 | 69.8927 | 66.5739 | 66.0769 |
> > > | 67.5227 | 86.1465 | 51.8991 | 51.9126 | 57.2931 | 64.6606 | 56.237  | 60.516  |
> > > | 66.3006 | 75.3647 | 76.3211 | 53.7599 | 49.5313 | 64.5966 | 50.0822 | 72.2078 |
> > > | 64.2738 | 71.4521 | 65.6601 | 69.9979 | 54.4297 | 68.7699 | 57.5515 | 72.5907 |
> > > | 65.579  | 76.4331 | 61.4387 | 60.3955 | 67.2487 | 68.347  | 58.3225 | 69.7591 |
> > > | 65.5418 | 77.0624 | 64.1304 | 64.0231 | 64.177  | 75.9494 | 62.0601 | 71.8547 |
> > > | 64.2998 | 70.5066 | 65.4372 | 62.2888 | 58.4881 | 70.5489 | 75.8536 | 79.3775 |
> > > | 55.2992 | 67.74   | 58.7447 | 61.2894 | 56.3704 | 64.934  | 65.4651 | 84.7681 |
> > >
> > > **Base**
> > > | Task1 | Task2 | Task3 | Task4 | Task5 | Task6 | Task7 | Task8 |
> > > | ----- | ----- | ----- | ----- | ----- | ----- | ----- | ----- |
> > > | 96.04 | 0     | -1    | -1    | -1    | -1    | -1    | -1    |
> > > | 0     | 98.24 | 0     | -1    | -1    | -1    | -1    | -1    |
> > > | 0     | 0     | 94.27 | 0     | -1    | -1    | -1    | -1    |
> > > | 0     | 0     | 0     | 88.95 | 0     | -1    | -1    | -1    |
> > > | 0     | 0     | 0     | 0     | 76.47 | 0     | -1    | -1    |
> > > | 0     | 0     | 0     | 0     | 0     | 96.7  | 0     | -1    |
> > > | 0     | 0     | 0     | 0     | 0     | 0     | 85.92 | 0     |
> > > | 0     | 0     | 0     | 0     | 0     | 0     | 0     | 90.43 |
> > >
> > >
> > > **Base + JLT**
> > > | Task1 | Task2 | Task3 | Task4 | Task5 | Task6 | Task7 | Task8 |
> > > | ----- | ----- | ----- | ----- | ----- | ----- | ----- | ----- |
> > > | 98.02 | 60.59 | -1    | -1    | -1    | -1    | -1    | -1    |
> > > | 99.01 | 99.41 | 73.44 | -1    | -1    | -1    | -1    | -1    |
> > > | 98.52 | 97.65 | 98.96 | 76.8  | -1    | -1    | -1    | -1    |
> > > | 99.01 | 98.24 | 98.96 | 99.45 | 78.08 | -1    | -1    | -1    |
> > > | 96.04 | 98.82 | 98.44 | 98.9  | 99.46 | 95.6  | -1    | -1    |
> > > | 98.02 | 99.41 | 97.4  | 100   | 99.46 | 98.9  | 87.32 | -1    |
> > > | 94.06 | 97.65 | 93.23 | 99.45 | 97.33 | 97.8  | 100   | 73.4  |
> > > | 96.54 | 99.41 | 97.4  | 98.9  | 97.86 | 98.9  | 100   | 100   |
> > >
> > > While the backbone maintains stable performance on previous tasks, the Base model without JLT forgets all previous tasks and cannot even predict the immediately following task. However, once JLT is added, the LM head is trained to align with the backbone’s representations, enabling the model to achieve performance comparable to what the backbone inherently provides. Importantly, prior work [1] has already demonstrated that a classifier can enhance performance beyond the raw representational quality of the backbone.
> > >
> > > [1] DIRECTPROBE: Studying Representations without Classifiers, NAACL 2021.

---

> > > > ### Author Response · Authors · 2025-11-28
> > > > **Thank you for your thoughtful feedback. Please find our additional response. [2/2]**
> > > >
> > > > **Further clarification on kNN vs. clustering**
> > > >
> > > > We appreciate the reviewer’s insistence on this point and agree that kNN and clustering share some generic sensitivities to the underlying feature geometry. Our claim is not that clustering is “problem-free,” but that, in the context of incremental learning, it is better aligned with how such models are actually trained and with the quantities we wish to evaluate, in ways that go beyond the generic pros/cons of nonparametric methods. We clarify this more explicitly below.
> > > >
> > > > Supervised training encourages class-wise clusters in representation space.
> > > > In standard supervised learning, the training objective (e.g., cross-entropy over class logits) explicitly encourages samples from the same class to have similar representations and to be separated from samples of other classes. In encoder-style models, this is typically realized through a classifier trained on top of the [CLS] token, and in decoder-style models through label tokens in the LM head. In both cases, the learning signal pushes representations toward class-wise clusters. Our clustering-based evaluation directly inspects whether such class-specific clusters indeed emerge in the backbone feature space, which is precisely what the supervised objective is trying to achieve.
> > > >
> > > > Incremental learning further strengthens the role of class/label tokens.
> > > > In incremental learning, each new task introduces additional classes, and the corresponding classifier weights, [CLS] representations, or label-token embeddings are updated over time. Good IL performance therefore presupposes that examples from the same class remain grouped together in the backbone representation space across tasks, rather than being collapsed or entangled with other classes as new tasks arrive. From this perspective, clustering is not an arbitrary alternative to kNN, but a natural way to test whether the representation still exhibits the class-wise grouping structure that IL training (via class and label tokens) is designed to create.
> > > >
> > > > A simple, test-time protocol based solely on backbone features.
> > > > Our evaluation protocol is intentionally simple and intuitive:
> > > >
> > > > (i) we pass the test set through the backbone to obtain features,
> > > >
> > > > (ii) we cluster these features into K clusters, where K equals the number of target classes, and
> > > >
> > > > (iii) we measure the alignment between clusters and ground-truth classes via ARI/NMI.
> > > >
> > > > Labels are used only in this final matching step; the grouping itself is driven purely by the feature geometry. This provides a direct view of how well the backbone alone organizes the data into class-like groups. In contrast, kNN always requires an explicit labeled reference set at test time, whose composition (how many exemplars per class, which memory budget, which sampling policy) is tightly entangled with the design of the incremental learning scenario.
> > > >
> > > > **Forward transfer to unseen classes** is fundamentally easier to capture with clustering than with kNN.
> > > > Forward transfer is specifically concerned with how well the current representation supports classes that have not yet been trained on. For such classes, a kNN classifier is conceptually problematic: by definition, kNN needs labeled reference examples for each class it predicts, but for unseen classes these labeled references are either unavailable (respecting the IL setting) or must be supplied from the future, which breaks the incremental learning assumption. This makes kNN a poor fit for evaluating FWT in a principled way.
> > > > Clustering, on the other hand, operates on unlabeled test features and only uses labels post hoc for evaluation. This allows us to meaningfully assess whether the backbone already tends to separate and group instances of unseen classes—even though the model has never received supervised signals for them. In our view, this is a substantive conceptual advantage of clustering over kNN in the IL + FWT setting, rather than a minor technical detail.
> > > >
> > > > Taken together, these points go beyond the generic observation that “all nonparametric methods have limitations.” Our choice of clustering is driven by how supervised and incremental learning actually shape the backbone representations (via class and label tokens), and by the need to evaluate both backward and forward transfer—particularly on unseen classes—without assuming access to labeled reference sets that would violate the IL protocol. We will revise the paper to make these IL-specific motivations more explicit, so that our preference for clustering over kNN is understood as a principled design choice rather than a purely aesthetic one.
> > > >
> > > > Once again, your insightful comments have significantly improved our work, and it was an honor to have the opportunity to discuss them with you. Thank you.
> > > >
> > > > Sincerely,
> > > > The Authors

---

### Official Review · Reviewer_oPJg · 2025-10-31

**Soundness:** 3
**Presentation:** 3
**Contribution:** 2
**Rating:** 6
**Confidence:** 3

**Summary:**

This paper investigates incremental learning in natural language processing. First, it conduct extensive studies to diagnose the insights of FWT and BWT on classifier, probing classifier and backbone. Beyond such empirical analysis, it develops a simple and effective method namely Just LM-Head Tuning (JLT), which can be applied to all IL methods. Thanks to JLT, existing IL methods achieve significant improvements on stability and plasticity.

**Strengths:**

The motivation on studing FWT and BWT is interesting and important for incremental learning.

This work conducts many empirical experiments to shed light on the differences between classifier, probing classifier and backbone.

The proposed JLT method demonstrates consistent and promising gains over existing baselines.

**Weaknesses:**

Apart from empirical analysis, this paper lacks theoritical evidence, to explain the  findings.

The motivation behind JLT is not totally clear. It is unknown about why LM can help overcome the problem in FWT and BWT.

The algorithm of JLT needs more explaination. In addition, how can it be integrated with other baselines. It is needed to provide a specific example.

Furthermore, it is helpful for understanding JLT given a new figure.

**Questions:**

In Figure 2, it claims there are 8 methods. However, how to tell the results from these methods?

In Figure 3 and 4, it lacks explaination on the results achieved by different methods.

This work conducts experiments with NLP benchmarks. It is interesting to clarify whether it can be applicable to computer vision benchmarks as well.

---

> ### Author Response · Authors · 2025-11-26
> **Response to the Concerns**
>
> Thank you very much for taking the time to review our work. We sincerely appreciate your thoughtful evaluation, especially your positive assessment of the aspects we aimed to highlight and contribute in the “Strength” section. Below are our responses to the weaknesses and questions you raised.
>
> **Weaknesses**
>
> We acknowledge that our motivation, reasoning, and explanations regarding JLT were not sufficiently detailed. We have provided similar clarifications to the other reviewers as well and have submitted an Official Comment addressing all reviewers. In addition, as you noted, we are preparing a revised version of the paper that includes improved figures and clearer explanations of JLT. Please refer to our Official Comment for further details.
>
> **Question**
>
> **Q1 & Q2**
>
> We acknowledge that there were parts of our presentation and explanation that were not sufficiently clear, and we kindly ask for your understanding. Below is our response.
>
> The eight IL methods mentioned in Figure 2 refer to the following methods listed in Section 3.1 Task & Baseline: Base, DERpp, CLSER, L2KD, LAMOL_g, LAMOL_t, LAMOL_KD, and PCLL.
>
> Figures 3 and 4 present results obtained from multiple models for each IL method. The reason we presented them in this format is that, although some tendencies may vary depending on the type or size of the model, our goal was to visually illustrate that there is a trade-off relationship between BWT and FWT across IL methods. We will revise the paper to include clearer explanations and captions for each figure.
>
>
> **Q3**
>
> Although we have already conducted experiments on computer vision benchmarks, we were unable to include them due to the extensive volume of results. The table below shows the Pearson correlations computed between BWT and FWT, using classifier, probing classifier, and backbone (NMI) for representative IL methods in the computer vision domain on CIFAR-10 with a ResNet-50 model. (We also have results for ResNet-34 and CIFAR-100.)
>
> | Method        | Cls BWT   | Cls FWT   | Prob BWT  | Prob FWT | Back BWT   | Back FWT   |
> |---------------|-----------|-----------|-----------|----------|-----------|-----------|
> | seq           | -84.4375  | 84.0375   | -42.7125  | 46.0125  | -23.5770  | 25.5622   |
> | er            | -81.1875  | 92.7500   | -19.2875  | 34.2000  | -19.5815  | 15.3585   |
> | er_ace_lider  | -21.8875  | 51.8750   | -15.3125  | 34.5875  | -17.9250  | 21.2757   |
> | derpp         | -47.0875  | 84.1250   | 32.3375   | 20.8250  | -17.4123  | 21.9662   |
> | derpp_lider   | -45.8750  | 87.5375   | -1.0500   | 31.7000  | -28.1063  | 31.3655   |
> | xder          | -33.4000  | 48.8000   | -11.8375  | 10.9875  | -15.6364  | 15.0344   |
> | **Pearson r** | **-0.7662** |           | **-0.6133** |          | **-0.8586** |           |
>
> What we can observe from the table closely mirrors our conclusions drawn in the natural language domain (particularly with generative models):
>
> 1. The strongest negative correlation between BWT and FWT appears in the backbone, and the stability–plasticity dilemma clearly holds in this setting.
> 2. In the probing classifier, the absolute magnitude of FWT is captured more prominently than that of BWT. This tendency is highly consistent with what we observed in Figure 4 (b) and (e).
> 3. Measurements obtained from the classifier were highly extreme across methods, showing trends that differ significantly from those observed in the backbone. This aligns closely with the main motivation of our study.
>
> In conclusion, the methods we proposed are also applicable to the computer vision domain, and the evaluation results support the assumptions underlying our work.
>
> Your suggestion allowed us to further verify the contributions our research can make. We sincerely appreciate your insightful comments. Please feel free to reach out if you have any additional questions.
>
> Thank you very much.
>
> The Authors.

---

### Author Response · Authors · 2025-11-14
**Justification Regarding Anonymity of the Supplementary Code Package**

First of all, we would like to express our sincere gratitude to the AC and all reviewers for taking the time to evaluate our work.

Reviewer X4ao pointed out that the README file in the supplementary code package contains information about the authors’ identities. We apologize for the confusion.

However, we would like to clarify that this information does not pertain to us at all; it is metadata from the baseline code, which has already been published in 2024. We have no connection whatsoever to that publication, and none of our personal information is included.

If this still constitutes a violation of the anonymity policy, we respectfully ask the AC to make a judgment. We believe it is best to clarify this matter before the rebuttal begins, and therefore kindly request your review.

Thank you very much.

Sincerely,
The Authors

---

> ### Comment · Reviewer_X4ao · 2025-11-14
>
> Thank you for your reply.
> This issue should be left to the AC for review, so the questions I raised about the paper itself and my rating did not involve this factor.
> Best regards.

---

> > ### Author Response · Authors · 2025-11-14
> >
> > Dear Reviewer,
> >
> > Thank you very much for checking this matter so promptly.
> > We asked the AC to verify this issue first because, if it were grounds for a desk rejection, we felt there would be no need to take up the valuable time of the reviewers.
> >
> > Thanks to your careful confirmation, we were able to address this point before starting the full review process, and we sincerely appreciate it once again.
> >
> > We fully understand that your review comments are separate from this issue, and we can see how thorough and insightful your review is. We are preparing a detailed and sincere response, and we will proceed with the rebuttal soon.
> >
> > Thank you very much for your valuable time and feedback.
> > Sincerely,
> > The Authors

---

### Author Response · Authors · 2025-11-24
**Clear explanation and differentiation of JLT**

We would like to express our sincere gratitude to all reviewers for taking the time to evaluate our paper. Among the points raised, we would like to separately provide clarification regarding the methodology, excluding those related to presentation and clarity.

Existing approaches that freeze the backbone and train a classifier have been previously proposed. While such methods (1) prevent catastrophic forgetting by freezing the model, they also (2) suffer from the limitation that the backbone cannot acquire new knowledge. Our proposed JLT is different from approaches that freeze the backbone after only part of the IL tasks.

1. JLT allows both the backbone and the LM head to continue learning throughout the entire IL process.
2. JLT uses the model’s original LM head and does not rely on any additional structures, such as a separate classifier.
3. After completing all IL steps, JLT performs a simple final step—retraining the LM head using the backbone’s representations—so that the LM head aligns with the fully trained backbone.

As a result, JLT offers the following advantages:

1. The backbone continues to acquire knowledge.
2. Because it follows a standard IL process, JLT can operate in a plug-and-play manner with other IL methods such as Replay, Knowledge Distillation, and Variational Autoencoders.
3. It does not require additional components, nor does it require extra training for each task.

The illustration in Figure 1 is not an overview of JLT, but rather an overview of the three evaluation methods. In the revised version, we will include an overview and detailed description of JLT to clearly distinguish it from existing methods.

We sincerely apologize for the lack of clarity in our original explanation, and we are truly grateful for the time and effort you have devoted to reviewing our work.

Sincerely,

The Authors

---

### Author Response · Authors · 2025-12-04
**Thank You for Your Efforts as the New AC [1/2]**

**Dear new AC,**

We would like to express our deep gratitude for your efforts as the newly assigned AC following the unfortunate incident at ICLR. We thoroughly enjoyed and were honored to engage in discussions grounded in the remarkable insights of the esteemed reviewers, and we regret that we could not continue those exchanges until the end.

We would like to clarify in advance that we do not know any personal information about the reviewers, and that through the normal rebuttal process, two reviewers raised their scores and provided additional positive indications. We also want to emphasize that we have received all reviewers’ comments with an open mind, responded with utmost humility and respect, and considered it a great honor to interact with all reviewers and the AC.

Below is a summary of the key contributions of our work and the main points addressed during the rebuttal process, provided to assist your evaluation.

The two major concerns that reviewers commonly raised were both fully acknowledged on our side. We addressed them in detail throughout the rebuttal process, and they can be summarized as follows:

**1. Improving the clarity of the JLT method**

The core idea of JLT is that “if the backbone already possesses sufficient capability, properly transferring that capability into the LM Head (classifier) is all that is needed.” Although many existing methods train an additional classifier, such approaches often freeze the backbone to prevent catastrophic forgetting—resulting in the unintended consequence of blocking the backbone from acquiring new knowledge.

In contrast, our JLT approach is fundamentally different:

1. It does not introduce a separate LM Head (classifier); instead, it directly utilizes the model’s original LM Head.
2. It never freezes the backbone throughout the entire IL process, allowing the backbone to experience both backward transfer (BWT) and forward transfer (FWT).
3. It performs the entire IL process without any freezing, additional structures, or auxiliary techniques, making it seamlessly plug-and-play with any IL method.

We explained these points thoroughly during the rebuttal, and in the final revision we added a dedicated figure and expanded descriptions for clarity.

**2. Improving the logical flow of the paper**

Although we conducted a wide variety of experiments, we humbly acknowledge the reviewers’ point that this breadth unintentionally obscured the core contributions. In response, we moved some less essential experiments (e.g., Model Size & Capacity analyses) to the Appendix and increased the emphasis on explaining JLT.

Based on the discussions with reviewers, we clarified the overall logic of our work in the revised paper as follows:

1. Based on classifier-based evaluation, IL methods have historically been presented as if catastrophic forgetting is largely solved.
2. However, these evaluations largely overlooked FWT. When evaluating at the backbone level using our proposed FWT metric, we discovered a trade-off between BWT and FWT.
3. Even under the base learning approach, the backbone neither forgot everything nor learned entirely from scratch; rather, it retained meaningful anti-forgetting and exhibited forward transfer.
4. Therefore, leveraging the backbone alone already provides sufficient backward and forward transfer, highlighting the importance of backbone-centric evaluation and methods such as JLT.

**3. Detailed explanation and examples for the three evaluation methods**

We evaluated IL-trained models using three different output:

1. The model’s original LM Head (classifier) output
2. An independent LM Head (classifier) output
3. The backbone’s representations

The differences across the three evaluation perspectives are shown below.

---

> ### Author Response · Authors · 2025-12-04
> **Thank You for Your Efforts as the New AC [2/2]**
>
> **Backbone (Spectral Algorithm NMI)**
>
> | Task1 | Task2 | Task3 | Task4 | Task5 | Task6 | Task7 | Task8 |
> | --- | --- | --- | --- | --- | --- | --- | --- |
> | 76.294 | 68.2249 | 62.995 | 49.5796 | 46.2606 | 69.8927 | 66.5739 | 66.0769 |
> | 67.5227 | 86.1465 | 51.8991 | 51.9126 | 57.2931 | 64.6606 | 56.237 | 60.516 |
> | 66.3006 | 75.3647 | 76.3211 | 53.7599 | 49.5313 | 64.5966 | 50.0822 | 72.2078 |
> | 64.2738 | 71.4521 | 65.6601 | 69.9979 | 54.4297 | 68.7699 | 57.5515 | 72.5907 |
> | 65.579 | 76.4331 | 61.4387 | 60.3955 | 67.2487 | 68.347 | 58.3225 | 69.7591 |
> | 65.5418 | 77.0624 | 64.1304 | 64.0231 | 64.177 | 75.9494 | 62.0601 | 71.8547 |
> | 64.2998 | 70.5066 | 65.4372 | 62.2888 | 58.4881 | 70.5489 | 75.8536 | 79.3775 |
> | 55.2992 | 67.74 | 58.7447 | 61.2894 | 56.3704 | 64.934 | 65.4651 | 84.7681 |
>
> ---
>
> **Base**
>
> | Task1 | Task2 | Task3 | Task4 | Task5 | Task6 | Task7 | Task8 |
> | --- | --- | --- | --- | --- | --- | --- | --- |
> | 96.04 | 0 | -1 | -1 | -1 | -1 | -1 | -1 |
> | 0 | 98.24 | 0 | -1 | -1 | -1 | -1 | -1 |
> | 0 | 0 | 94.27 | 0 | -1 | -1 | -1 | -1 |
> | 0 | 0 | 0 | 88.95 | 0 | -1 | -1 | -1 |
> | 0 | 0 | 0 | 0 | 76.47 | 0 | -1 | -1 |
> | 0 | 0 | 0 | 0 | 0 | 96.7 | 0 | -1 |
> | 0 | 0 | 0 | 0 | 0 | 0 | 85.92 | 0 |
> | 0 | 0 | 0 | 0 | 0 | 0 | 0 | 90.43 |
>
> ---
>
> **Base + JLT**
>
> | Task1 | Task2 | Task3 | Task4 | Task5 | Task6 | Task7 | Task8 |
> | --- | --- | --- | --- | --- | --- | --- | --- |
> | 98.02 | 60.59 | -1 | -1 | -1 | -1 | -1 | -1 |
> | 99.01 | 99.41 | 73.44 | -1 | -1 | -1 | -1 | -1 |
> | 98.52 | 97.65 | 98.96 | 76.8 | -1 | -1 | -1 | -1 |
> | 99.01 | 98.24 | 98.96 | 99.45 | 78.08 | -1 | -1 | -1 |
> | 96.04 | 98.82 | 98.44 | 98.9 | 99.46 | 95.6 | -1 | -1 |
> | 98.02 | 99.41 | 97.4 | 100 | 99.46 | 98.9 | 87.32 | -1 |
> | 94.06 | 97.65 | 93.23 | 99.45 | 97.33 | 97.8 | 100 | 73.4 |
> | 96.54 | 99.41 | 97.4 | 98.9 | 97.86 | 98.9 | 100 | 100 |
>
> ---
>
> These results show that even when undergoing the exact same IL process, the backbone and LM Head behave in fundamentally different ways. To assist the very busy AC, we included these qualitative examples to illustrate the distinctions clearly.
>
> We made substantial efforts to improve clarity and presentation. Beyond these refinements, we believe that the core novelty and contributions of the work were acknowledged by all reviewers. Although the productive rebuttal process and the resulting score increases from two reviewers were unfortunately nullified—and we were deprived of the opportunity for further positive updates—we trust that the AC will carefully review the rebuttal exchanges and reach a wise decision.
>
> We extend our sincere gratitude for the AC’s unexpected additional workload, and we offer our deepest appreciation to all reviewers who engaged with us in the discussion.
>
> With sincere respect,
>
> The Authors

---

### Meta-Review · Area_Chair_tyCk · 2026-01-05

**Summary:**

The paper studies the problem "Incremental Learning (IL)", which has long been an important research area in neural networks. The paper is motivated by recent studies which have demonstrated that the backbone exhibits sufficiently strong anti-forgetting capabilities, while the classifier is the primary source of forgetting. They re-establish the   metrics BWT (Backward Transfer) and FWT (Forward
Transfer) and analyze the correlation between the two. Moreover, the paper proposes the method called "Just LM-Head Tuning (JLT)",  to leverage the backbone trained through the IL process to transfer the LM Head.

Three reviewers have commented on this paper, and posed several concerns regarding various aspects, like the algorithm, motivation, presentation, and some experimental details.

**Reviewer Concerns:**

The authors provide detailed responses to the questions and concerns, and I think some of them have been addressed. However, my overall feeling is that the proposed method is very heuristic and engineering, and lack of theoretical analysis in depth. Given the high bar of ICLR, I don't think the current manuscript is sufficiently good for acceptance.

**Reviewer Scores:**

Initially, the three reviewers give the scores 6, 2, 2. During the discussion, Reviewer JJQR and Reviewer X4ao are willing to increase their scores, but the average assessment is still negative.

---

### Decision · Program_Chairs · 2026-01-26

Reject